# Thermokarst lake inception and development in syngenetic ice-wedge polygon terrain during a cooling climatic trend, Bylot Island (Nunavut), eastern Canadian Arctic

Frédéric Bouchard[1,2], Daniel Fortier[2,3], Michel Paquette[4], Vincent Boucher[5], Reinhard Pienitz[2,5], Isabelle Laurion[2,6]

[1] Géosciences Paris Sud (GEOPS), Université Paris Saclay, Orsay, France
[2] Centre d'études nordiques (CEN), Université Laval, Québec, Canada
[3] Département de géographie, Université de Montréal, Montréal, Canada
[4] Department of Geography and Planning, Queen's University, Kingston, Canada
[5] Département de géographie, Université Laval, Québec, Canada
[6] Centre Eau Terre Environnement, Institut national de la recherche scientifique (INRS-ETE), Québec, Canada

*Correspondence to*: Frédéric Bouchard (frederic.bouchard@u-psud.fr)

**Abstract.** Thermokarst lakes are widespread and diverse across permafrost regions and they are considered significant contributors to global greenhouse gas emissions. Paleoenvironmental reconstructions documenting the inception and development of these ecologically important water bodies are generally limited to Pleistocene-age permafrost deposits of Siberia, Alaska, and the western Canadian Arctic. Here we present the gradual transition from syngenetic ice-wedge polygon terrain to a thermokarst lake in Holocene sediments of the eastern Canadian Arctic. We combine geomorphological surveys with paleolimnological reconstructions from sediment cores in an effort to characterize local landscape evolution from a terrestrial to freshwater environment. Located on an ice-rich and organic-rich polygonal terrace, the studied lake is now evolving through active thermokarst, as revealed by subsiding and eroding shores, and was likely created by water pooling within a pre-existing topographic depression. Organic sedimentation in the valley started during the mid-Holocene, as documented by the oldest organic debris found at the base of one sediment core and dated at 4.8 kyr BP. Local sedimentation dynamics were initially controlled by fluctuations in wind activity, local moisture and vegetation growth/accumulation, as shown by alternating loess (silt) and peat layers. Fossil diatom assemblages were likewise influenced by local hydro-climatic conditions and reflect a broad range of substrates available in the past (both terrestrial and aquatic). Such conditions likely prevailed until ~ 2000 BP, when peat accumulation stopped as water ponded the surface of degrading ice-wedge polygons, and the basin progressively developed into a thermokarst lake. Interestingly, this happened in the middle of the Neoglacial cooling period, likely under colder-than-present, but wetter-than-average, conditions. Thereafter, the lake continued to develop as evidenced by the dominance of aquatic (both benthic and planktonic) diatom taxa in organic-rich lacustrine muds. Based on these interpretations, we present a four-stage conceptual model of thermokarst lake development during the late Holocene, including some potential future trajectories. Such a model could be applied to other formerly glaciated syngenetic permafrost landscapes.

# 1 Introduction

Lakes are extremely abundant across the circumpolar regions, with several millions of waterbodies spread over an estimated total surface area ranging from ~ 1.4 to 1.8 x $10^6$ km$^2$ (Muster et al., 2017; Paltan et al., 2015; Verpoorter et al., 2014). The vast majority of these aquatic systems are located in permafrost environments, especially in lowland regions with moderate to high excess ground-ice content (typically > 30% in volume) and a thick sediment cover (Grosse et al., 2013; Smith et al., 2007). Recent high-resolution mapping efforts reported significant variability in waterbody size distributions across permafrost regions (Muster et al., 2017). Thermokarst (thaw) lakes occur mainly in ice-rich permafrost regions, where ground-ice melting can result in localized ground surface subsidence, water accumulation, and self-maintained lake expansion (van Everdingen, 1998). These lakes vary greatly in morphology, depth (< 1 m to several meters deep in most cases) and area (from a few meters across to several km$^2$) depending on ground-ice content and distribution, lake age, hydro-climatic conditions and local topography (e.g., Côté and Burn, 2002; Hopkins, 1949; Pienitz et al., 2008). Some of the lakes located in unglaciated ice-rich (Yedoma) terrains of Siberia, Alaska and western Canada started to develop in the late Pleistocene and form a separate lake category, up to several tens of meters deep (e.g., Farquharson et al., 2016; Lenz et al., 2016;). However, the majority of thermokarst lakes across the Arctic are shallow (a few meters) and most of them were formed in formerly glaciated terrains during the Holocene (Grosse et al., 2013; Smith et al., 2007).

Thermokarst lake evolution involves a remarkably diverse suite of hydro-climatic, geomorphological and ecological processes (Bouchard et al., 2017; Grosse et al., 2013). Although modern thermokarst processes and landforms may involve anthropogenic causes, thermokarst development during the Holocene can be associated to three main drivers: increased air and ground temperatures, ground disturbances (through fluvial, thermal or ecological mechanisms, e.g. slumps or fires), and snow accumulation (e.g., Anderson et al., 2019; French, 2017). In the latter case, the insulating capacity of a thick snow cover can substancially prevent ground cooling during the winter, resulting in significantly higher ground temperatures (near 0°C), hence promoting localized or widespread permafrost thawing the following summer (Frauenfeld et al., 2004; Morse et al., 2012). When lake depth exceeds the maximum thickness of winter ice cover, bottom water stays unfrozen throughout the year and mean annual lake-bottom temperature remains above 0 °C, resulting in the formation of a talik (thaw bulb) underneath the lake (Burn, 2002). Once initiated, thermokarst lakes in continuous permafrost tend to develop laterally; first by the coalescence of polygonal and/or ice-wedge trough pools overlying melting ice-wedge networks (Czudek and Demek, 1970; Mackay, 2000; French, 2017), and then by thermal and mechanical shoreline erosional processes, such as wave-induced erosional niche development or mass wasting through thaw slumping and block failures (Kokelj and Jorgenson, 2013). Ultimately, and depending on local landscape conditions (e.g., soil type, vegetation cover, topography), thermokarst lake development generally ends with one or more of the following: rapid drainage resulting from shoreline breaching, either during higher-than-average lake-level episodes (e.g., Jones and Arp, 2015; Lantz and Turner, 2015; Mackay and Burn, 2002; Turner et al., 2010) or due to ice wedge melting and thermal erosion gullying (e.g., Fortier et al., 2007; Godin and Fortier, 2012); lake-level drawdown due to factors that lead to increased evaporation (Bouchard et al., 2013a; Riordan et al., 2006); subsurface drainage

(groundwater infiltration) through an open talik (Yoshikawa and Hinzman, 2003); or terrestrialization via rapid peat accumulation and lake infilling (Payette et al., 2004; Roach et al., 2011).

Thermokarst lakes play a key role in the global carbon cycle (e.g., Cole et al., 1994; Serikova et al., 2019; Wik et al., 2016), because they form in areas where organic carbon is stored in frozen soils (Hugelius et al., 2014; Schuur et al., 2015).

Consequently, they are biogeochemical hotspots through their release of substantial amounts of carbon dioxide ($CO_2$) and methane ($CH_4$) to the atmosphere (e.g., Abnizova et al., 2012; Laurion et al., 2010; Matveev et al., 2018; Walter et al., 2007). A fundamental aspect of thermokarst lakes is the age (millennium-old *vs*. modern) of the carbon stored in the frozen soil and released by thermokarst ecosystems, which is linked to the potential of a thermokarst lake to generate a positive feedback on climate (Elder et al., 2018; Mann et al., 2015; Vonk et al., 2013). Carbon older than ~ 500 to 1000 years can be considered as

'in excess' in the system, thus representing a net atmospheric contribution from a formerly stable reservoir (Archer et al., 2009). Work conducted in the eastern Canadian Arctic, in eastern Siberia and in Alaska has shown that radiocarbon age can indeed vary by several orders of magnitude over a small area depending on waterbody properties (Bouchard et al., 2015a; Dean et al., 2020; Elder et al., 2018). Yet, the majority of studies focusing on the age and sources of $CO_2$ and $CH_4$ released by thermokarst lakes come from Yedoma regions, which represent a small fraction (~ 4-6 %) of total permafrost areas (~ 1.0-1.4

x$10^6$ km$^2$ out of 23 x$10^6$ km$^2$ in total; e.g., Strauss et al., 2017). Lakes formed in formerly glaciated terrains are widespread across the Arctic and can contribute significantly to global greenhouse gas emissions (Smith et al., 2007; Wik et al., 2016).

Here we document the development of a thermokarst lake in a tundra valley of the eastern Canadian Arctic (Bylot Island, Nunavut, Canada) during the Holocene. We test the hypothesis that this lake developed following local landscape dynamics, and not solely because of increased temperatures. The lake is located within an ice-rich and organic-rich syngenetic permafrost

environment (Fortier and Allard, 2004). It thus serves as an interesting case-study of a landscape that is under-represented in the thermokarst literature. We combine high-resolution lake mapping, geomorphological observations and paleolimnological reconstructions (both litho- and biostratigraphy) in an effort to 1) document the inception and evolution of a thermokarst lake in syngenetic ice-wedge polygon terrain, 2) characterize the transition from terrestrial to aquatic conditions in a tundra valley setting, and 3) present a conceptual model of thermokarst development in syngenetic ice-wedge polygon terrain during a cool

climate episode of the late Holocene.

## 2 Study site

Bylot Island (Nunavut) is located in the Eastern Canadian Arctic, within the continuous permafrost zone (Fig. 1a). Most of the island is mountainous, and several glaciers spread from its center to peripheral lowland areas (Fig. 1b). The valleys of these glaciers were shaped during the successive Pleistocene glaciations (Klassen, 1993), and since the Holocene they developed

into highly dynamic biogeosystems rich in vegetation, ground ice, peat, and aquatic environments (Allard, 1996; Fortier and Allard, 2004). The prevailing climate is polar with a slight marine influence. Based on the 1980-2010 climate normals from the closest meteorological station located ~ 80 km away in the village of Mittimatalik (Pond Inlet) on Baffin Island (72° 41'

N; 77° 58' W), the mean annual air temperature is -14.6 °C, with average daily temperatures ranging from -33.4 °C in January to 6.6 °C in July, and a total precipitation of 189 mm, of which 91 mm fall as rain between June and September (Environment Canada, 2019). Thawing and freezing degree-days are around 475 and 5735, respectively. Winter, defined here as when continuous daily mean air temperature remains < 0° C, lasts from early September to mid-June, for an average total of 283 days per year. A station operated since 2004 by the Center for Northern Studies (CEN) at the study site provides similar climate data (CEN, 2018).

The study site (73° 09' N; 79° 58' W) is located in the valley locally named Qarlikturvik, which has a NE-SW orientation and a surface area of ~ 65 km$^2$ (~ 15 km-long x 4-5 km-wide) (Fig. 1c). A terminal moraine, located roughly halfway between the actual glacier (C-79) front and the seashore and sitting on marine clay, was $^{14}$C-dated to ~ 9.8 kyr BP (Allard, 1996). Holocene glacial retreat was accompanied by a marine transgression phase, which ended around 6 kyr BP (Allard, 1996). Like the majority of glaciers on Bylot Island, the C-79 glacier has recently been retreating from 0.9 to 1.8 km up the valley since the early 20$^{th}$ century, with most retreat occurring between 1958/1961 and 2001 (Dowdeswell et al., 2007). Marine clays deposited during the postglacial transgression phase were subsequently covered by glacio-fluvial sands and gravels (Fortier and Allard, 2004). Today, a braided river flows through the glacio-fluvial outwash plain, carrying sediments towards a delta aggrading in Navy Board Inlet. This outwash plain is bordered on both sides by a 3- to 5-m thick terrace, crisscrossed by networks of tundra polygons associated with the formation of syngenetic ice wedges. Along the southern bank of the river, the upper portion of this terrace is composed of alternating mineral (wind-blown sand and silt) and organic (peat) material, which started to accumulate over glacio-fluvial sands and gravels at least 3700 years ago (Fortier and Allard, 2004). These peaty loess deposits in which permafrost aggrades syngenetically are typically ice-rich, and their organic matter content can reach over 50 %. The active layer depth in such deposits generally ranges between 40 to 80 cm (down to 1 m in sandy/gravelly material), and the maximum depth of permafrost on Bylot Island has been estimated to be over 400 m (Allard et al., 2016; Smith and Burgess, 2000).

The sampled lake, informally named Gull Lake (maximum depth ~ 4.2 m), is located within the lake- and pond-rich polygonal terrace, near the terminal moraine (Fig. 1c). Limnological observations conducted during the ice-free season indicate relatively low concentrations of dissolved organic carbon (DOC), nutrients and ions in Gull Lake compared to the surrounding ice-wedge troughs and coalescent polygonal ponds, as well as a thermally homogenous and well-oxygenated water column. However, dissolved oxygen concentrations decrease rapidly under the winter ice cover and at the bottom of the lake, near the water-sediment interface (Bouchard et al., 2015a). Greenhouse gas (GHG) sampling and dating showed that this lake is a relatively small but spatially variable source of dissolved and ebullition GHG, with millennium-age methane released in its center (up to 3.5 kyr BP) and peripheral shallow zones (up to 2.8 kyr BP). The age of methane emitted from the central zone is almost corresponding to the maximum age of syngenetic organic sedimentation in the valley (3.7 kyr BP) (Bouchard et al., 2015a; Fortier and Allard, 2004).

# 3 Materials and methods

## 3.1 Lake watershed and geomorphology

A portable sonar system, equipped with an internal GPS antenna (Humminbird model 859XD) and mounted on a small zodiac, was used to map the lake bottom in July 2014 (as in Bouchard et al., 2015b). Lake-depth signals were continuously recorded along regularly spaced (20-25 m apart) navigation lines, mainly of SW-NE and SE-NW orientations. Depth data were interpolated between navigation lines using the compatible software (AutoChart) to produce a geo-referenced 3D bathymetric map. The acquired data were also used to calibrate and extrapolate lake bottom depths inferred from GPR mapping conducted the following year.

Ground penetrating radar (GPR) surveying on lake ice cover allows accurate description of lake bottom topography (Moorman, 2001; Paquette et al., 2015; You et al., 2017).  Three GPR survey lines crossing the lake were done in May 2015 (Fig. 2) using a sleigh-dragged Sensors and Software PulseEkko GPR and 50 MHz antennas. GPR line processing was performed using Ekko project software and included Dewow, Lowpass temporal filter to diminish background noise, and a background average subtraction to remove the overwhelming ice/water boundary signal. The base of the ice cover and lake bottom depth were manually identified and corresponded well to the dielectric properties of ice and water. Signal travel velocities of 0.06 m ns$^{-1}$ and 0.13 m ns$^{-1}$ were used respectively for unfrozen and frozen ground, and GPR vertical signal resolution (no pulse overlap) is approximately 1 m, but slightly lower in ice and frozen ground than in water. There are important limiting factors affecting GPR signal at the bottom of a lake; the strong reflectivity of the ice-water interface (-0.67) and of the water-sediment interface (+0.5), as well as the high permittivity of water (80) quickly diminish the signal intensity.

Temperature in surface sediments (near bottom waters) was monitored over a full year (July 2014 to July 2015) at 1-hour intervals using thermal sensors (Hobo U12; accuracy ± 0.25 °C; resolution 0.025 °C; operation range -40 to 125 °C) deployed at two sites: 1) near the lake center, in deeper waters (> 4 m) and 2) in the shallow peripheral zone (see section 4.1 below).

## 3.2 Sediment core sampling and logging

Sediment cores were collected from the same location (Fig. 2) during two consecutive years: 1) a shorter core (54 cm) from a boat during the ice-free season of 2014 (July), and 2) a longer core (109 cm) from the ice cover in spring 2015 (June). The coring site location (> 4 m water depth) was located in 2014 and 2015 using the bathymetric data from the sonar and GPR surveys (Bouchard et al., 2015b). Each core was retrieved using a handheld percussion corer equipped with a 7-cm diameter clear polycarbonate tube (Aquatic Research Instruments). Coring was stopped when the tube would not penetrate further into the sediments (coarse sands and gravels). The 2014 core was subsampled in the field immediately after retrieval at 1-cm intervals, and the subsamples were transferred into polyethylene bags and brought back to the laboratory where they were kept in the dark at 4 °C. For the 2015 core, water from above the sediment surface was removed immediately after retrieval in order to minimize the mixing of the water-sediment interface (Bouchard et al., 2011). The core was then stored vertically at non-freezing conditions for at least 48 h, allowing the upper sediments to slowly consolidate by dewatering, after which the

supernatant was removed. It was finally sealed with foam blocks to minimize potential disturbances during subsequent transport, brought back to the laboratory and stored in the dark at 4 °C for further analyses.

The 2014 core was visually examined in the field before and during subsampling to identify general stratigraphic units, such as gyttja (organic-rich lacustrine mud), peat, silt and sand, whereas the 2015 core was described with more detail in the laboratory. First, a non-destructive computed tomographic scan was performed to visualize internal sedimentary structures and infer sediment density (Supplement S1) (Calmels and Allard, 2004). This core was then cut into halves along its longitudinal axis with a rotating saw. One half was covered with a plastic film to minimize surface oxidation and desiccation and archived in the dark at 4 °C, and the other half was used for stratigraphic descriptions. Subsampling was then performed on the working half at 1-cm intervals, and the subsamples were freeze-dried (at least 48 h, depending on water content) and transferred into polyethylene bags for further analyses.

### 3.3 Lithological and chronological analyses

Sediment subsamples from the 2014 and 2015 cores were used to perform physical and chronological analyses. About 0.5 g of dry sediment was extracted to perform loss-on-ignition (LOI) measurements by a subsequent combustion at 550 °C for 4 h (Heiri et al., 2001). Organic matter content (LOI) and wet/dry sediment mass measurements were used to determine the sediment dry bulk density and to correlate both cores, in addition to visual descriptions. Supplementary subsamples were used to perform grain-size distribution analysis by sieving material coarser than 62.5 μm (i.e. sand and gravel) (ASTM, 2004) and using hydrometry for fine sediments (i.e. silt and clay) (ASTM, 2017).

Bulk sediment samples and, when present, fossil organic/wood fragments were carefully extracted and dried in glass bottles at 105 °C (Björck and Wohlfarth, 2001). Samples were pre-treated (HCl-NaOH-HCl) and combusted to $CO_2$ at the Radiocarbon Dating Laboratory (Université Laval, Québec QC, Canada) and $^{14}C$ dated by accelerator mass spectrometry (AMS) at Keck Carbon Cycle AMS Facility (University of California, Irvine CA, USA). Radiocarbon dates were reported using Libby's half-life (5568 yr), corrected for natural fractionation ($\delta^{13}C = -25$ ‰ PDB), and calibrated with the CALIB 7.1 online program (Stuiver et al., 2019) using the IntCal13 calibration data set (Reimer et al., 2013). Supplementary $^{14}C$ dates available for the surrounding frozen peat deposits, some of them not yet published, were also compiled and added to the dataset (Allard, 1996; Ellis and Rochefort, 2004; 2006; Fortier et al., 2006; 2020).

### 3.4 Diatom analysis

Fossil diatom analysis was conducted at the Aquatic Paleoecology Laboratory of CEN on 60 subsamples from the 2015 core (each cm for the top 12 cm, then each 2 cm towards core bottom) following Bouchard et al. (2013b). Diatom valves were extracted from ~ 50-mg samples using acid ($H_2SO_4$-$HNO_3$) digestion techniques and mounted on microscope slides using *Naphrax*, a highly refractive resin (Battarbee et al., 2001). For each subsample, an average of 400 diatom valves were counted along transects using a Leica DMRX light microscope. Identification was carried out to the lowest taxonomic level possible (i.e., species or variety/morphotype) at 1000 × magnification. Taxonomic identification mainly followed Antoniades et al.

(2008; 2009), Fallu et al. (2000), Krammer (2000, 2002), Krammer and Lange-Bertalot (1986; 1988; 1991a; 1991b), Lavoie et al. (2008), and Zimmermann et al. (2010). The complete diatom dataset is available in open access (Pienitz et al., 2019). Diatom taxa representing at least 1% (relative abundance) in at least one sample were displayed on abundance diagrams using the C2 software (Juggins, 2014). Photos of most of these taxa were taken with a Leica DFC490 camera (mounted on the microscope) and were used to prepare plates of the representative taxa (Supplement S2). In the following sections, taxa names are presented as they appeared originally in consulted floras.

## 4 Results

### 4.1 Lake basin morphology

Gull Lake has an irregular shape and bathymetry. It is mostly SW-NE elongated, parallel to the main valley axis, with a maximum length and width of ~ 500 and 250 m, respectively, for a total surface of 116 x $10^3$ $m^2$. There is no apparent inlet; however, the lake receives influx of snowmelt water during the spring. A small outlet, draining towards a nearby lake and the proglacial river to the North, is observable along the northern shore (Figs. 1 and 2). Based on the bathymetric map and satellite imagery, the lake can be separated into two zones: a shallow platform and a deeper basin. The shallow platform (< 2 m deep) occupies the periphery of the lake while gently dipping towards the lake center. Submerged ice-wedge polygons can be seen on this platform, as well as degraded furrows observed during the ice-free period with a submersible camera (Bouchard et al., 2015b; Video Supplement VS1). This morphology confirms that the lake is currently evolving through lateral thermokarst encroachment. The deeper (~ 2-4.2 m deep) basin occupies the center of the lake. This area is relatively bumpy, with shallower zones that can be distinguished from a boat or on the satellite image (Fig. 2). The central basin appears asymmetrical, with maximum depths concentrated within the SW portion of the lake.

GPR surveys conducted on top of lake ice during spring (early June) 2015 provide further information about Gull Lake morphology and winter conditions (Fig. 3) (open access data are in Fortier et al., 2019). Lake ice thickness averages 2.1 m +/- 0.1 m in the central basin, i.e. in areas where ice is not grounded. Lake depth is typically deeper than 3.2 m on average within the central basin. In contrast, mean depth is < 1 m within the peripheral platform, where the lake ice reaches the bottom and the freezing front penetrates lake sediments. Apart from the slightly steeper slope (6° on average) between the shallow platform and the deeper central basin, local gradients are gentle and rarely exceed 4°. Finally, many strong electromagnetic reflectors are present in the ground on all GPR lines (Fig. 3). A first series of these reflectors are located underneath the lake bottom, in both shallow and deeper zones, at an average depth of 0.49 m under the sediment surface (range 0-1.53 m). Another group of deeper reflectors are visible only under the shallow peripheral platform, at a depth of 2.6 +/- 1.4 m. The signal velocity (> 0.13 m $ns^{-1}$) based on the shape of some hyperbolas suggests that they occur in frozen material. All of these reflectors are located from ~ 5 to 40 m apart (apparent distance along GPR lines). Their occurrence at shallow depths beneath the central lake basin suggests that the lake does not have a deep thawed zone (talik) as is often the case underneath deep water bodies. However,

the temperature sensor installed at the bottom of the central basin indicates that surface sediments remain slightly above freezing conditions (1-2 °C) during nearly 9 months of the year (Supplement S3).

## 4.2 Lake sediment stratigraphy

### 4.2.1 Lithostratigraphy

Based on the description of the 2015 core, sedimentary units or zones appear as follows (Fig. 4) (Fortier and Bouchard, 2019a;
2019b), from bottom to top (the corresponding lithostratigraphy in the 2014 core, when observed, is mentioned at the end of each paragraph):

- **Lithozone 1** (109-80 cm). This unit is composed of mostly sand and gravel (> 50 %) with scattered peat and organic debris. Subzones 1a (109-103 cm) and 1c (86-80 cm) contain only sand and gravel, whereas subzone 1b (103-86 cm) contains organic debris, mostly in the form of cm-scale pieces of agglomerated peat. Compared to other units,
lithozone 1 has a relatively high mean density (~ 2 g cm$^{-3}$), typical of dominantly mineral material. Water content (20-40 %) and LOI (< 10 %) are relatively low, except for the above-mentioned peat and organic debris, as shown for example by a peak at 93-94 cm with 60 % water content and 25 % LOI. Based on $^{14}$C dating of one subsample (107-108 cm), this unit contains organic matter older than 5500 cal. yr BP (4805 $^{14}$C BP) (Table 1). No equivalent was found in the shorter 2014 core.

- **Lithozone 2** (80-10 cm). This unit is composed of medium to dark brown porous peat, moderately decomposed, interbedded with mm- to cm-thick silt and sand laminations. These silt/sand laminations are generally thicker (> 1 cm) and more present at the base of the unit than compared to the top. The average proportion of silt *vs*. sand in the mineral fraction is around 70 % *vs*. 30 %, respectively. An intermediate subzone 2b (55-35 cm), richer in sand (~ 50 %) and marked by convoluted horizons, separates subzones 2a (80-55 cm) and 2c (35-10 cm), which are both
dominated by peat. From the bottom to the top, there is a generally decreasing trend in density, from ~ 2 g cm$^{-3}$ (mostly mineral) to ~ 1 g cm$^{-3}$ (mostly organic material), with the exception of the above-mentioned subzone 2b. Meanwhile, there is an upward increase in water content (from 20 to > 60 %) and LOI (from < 10 to > 20 %), again with the exception of subzone 2b. This unit was observed in the 2014 core, at depths between 54 and 10 cm.

- **Lithozone 3** (10-0 cm). This unit is composed of laminated dark organic lacustrine mud (gyttja) overlying an organic-
poor silt layer (10 cm deep). The relative proportion of silt *vs*. sand in the mineral fraction is higher (> 80 % vs. < 10 %) compared to the underlying unit. With the exception of this silty mineral layer, the density is relatively low (~ 1.25 g cm$^{-3}$), typical of organic material. It has a high water content (60 to 80 %), similar to subzone 2c, but a medium LOI (~ 15 %), except for the basal silt layer (< 10 %). The basal silty layer was $^{14}$C-dated in both 2014 and 2015 cores (Table 1), yielding an age of around 2000 cal. yr BP (~ 2100 $^{14}$C BP). Similar sediments were observed in the 2014
core at the same depth (10-0 cm).

**4.2.2 Biostratigraphy**

A total of 230 diatom taxa (species or species groups) belonging to 52 genera were identified within the 60 thin sections prepared from the 2015 core (Pienitz et al., 2019). The average number of taxa for a given level was 43, ranging from a minimum of 5 (108-109 cm) to a maximum of 60 taxa (56-58 cm and 18-20 cm). Among these, the 15 most frequently
encountered taxa representing more than 5 % in relative abundance in at least one sample were selected to show major ecological changes that occurred in the past (Fig. 5). These changes were used to delimit diatom zones (or "biozones"), which are similar to the sedimentary units (or "lithozones") described above, although exact upper and lower limits are slightly different. These major biozones are as follows, from bottom to top:

- **Biozone 1** (109-74 cm). Compared to the entire core, this unit is characterized by a poor diversity in major taxa (n <
10) and in total counted taxa (n < 30 in average per level). The diversity is especially low in subzone 1a (109-102 cm), with only 5 major taxa and an average of 13 counted taxa per level. Notably, the *Diploneis-Geissleria* group, practically not observed anywhere else along the core, is overwhelmingly dominant within this subzone (> 20% of relative abundance). These species are generally associated with cold, oligotrophic, organic-poor, low conductivity and mostly alkaline (pH = ~ 8) waters, typical of Arctic streams and wetland headwaters (Antoniades et al., 2008;
Zimmermann et al., 2010). The overlying subzones 1b (102-86 cm) and 1c (86-74 cm) are notably more diverse in identified taxa (average total counted taxa of 33 and 34, respectively) and dominated by aerophilous/moss-associated species (e.g., *Chamaepinnularia soehrensis*, *Pinnularia sinistra*, *Diatomella balfouriana*), generally living in circumneutral to slightly acidic waters, typical of high-latitude peatlands (D. Antoniades, pers. comm.; Zimmermann et al., 2010). Abundant organic debris, in the form of cm-scale pieces of peat, were indeed observed in this unit (Fig.
4).

- **Biozone 2** (74-12 cm). This unit marks an increase in the abundance of major taxa (n >10), with the appearance of mostly small, benthic diatom genera (e.g., *Cavinula*, *Achnanthidium*, *Fragilaria*, *Staurosirella*) typical of shallow tundra ponds in ice-wedge polygon terrains, with cold waters and long-lasting ice cover (Antoniades et al., 2008; Ellis et al., 2008; Pienitz et al., 1995; Zimmermann et al., 2010). Total counted taxa are also much higher in this unit (nearly
50 on average per level), except in subzone 2b (48-34 cm) where the number of identified species per level ranges around 35. This intermediate subzone corresponds to the convoluted silt/sand horizons of lithozone 2 (subzone 2b; Fig. 4) and is mostly dominated by epiphytic, moss-associated genera (*Encyonema*, *Eunotia*, *Caloneis*) (Antoniades et al., 2008; Ellis et al., 2008; Zimmermann et al., 2010).

- **Biozone 3** (12-0 cm). This unit is similar to the underlying biozone 2, with a slightly higher number of major taxa
(n = 13) although a slightly lower total number of counted taxa (n = 47 on average per level). Thin sections were more concentrated in diatom valves within this zone, especially in the upper part (7-0 cm). Moreover, several taxa with a generally wide geographic distribution in lakes and wetlands and preferring high-nutrient waters (e.g., *Cavinula cocconeiformis*, *Cymbopleura naviculiformis*, *Eunotia bilunaris*) (Guiry and Guiry, 2019), not observed in other

zones, were counted within this unit (Pienitz et al., 2019). This biozone is the equivalent of lithozone 1 (laminated lacustrine mud; Fig. 4).

## 5 Discussion

Combining geomorphological and paleolimnological observations of Gull Lake basin and bottom sediments, we can reconstruct landscape dynamics in Qarlikturvik valley and lake development during the second half of the Holocene. This evolution was, however, strongly controlled by the early Holocene deglaciation of the valley, for which data from earlier studies are available. Hence, we first adopt a chronological approach in this section, covering the entire Holocene, in order to better 'set the stage' for Gull Lake's inception. We then present a four-stage conceptual model for thermokarst lake development in syngenetic permafrost of formerly glaciated terrains. Finally, we discuss the implications of some of our results for carbon dynamics in the Arctic.

### 5.1 Holocene history of the Qarlikturvik valley and ground-ice development

At the beginning of the Holocene, the Qarlikturvik valley was likely occupied by an inland-based glacier with a front advancing into shallow marine waters (Allard, 1996). This interpretation is based on the presence of marine shells (*Mya truncata* species) [14]C-dated at 9860 yr BP, and found within ice-contact deposits (sands, gravels and pebbles) with lithological properties corresponding to the surrounding Precambrian and Cretaceous-Tertiary rocks. Lacelle et al. (2018) and Coulombe et al. (2019) later proposed that Laurentide ice and Bylot ice were converging in the valley. Glacial retreat was then accompanied by a marine transgression phase, associated with the deposition of silts and clays, and which lasted until about 6000 yr BP ([14]C ages ranging from 9860 to 6100 yr BP in shells at different altitudes). Such fossiliferous marine sediments were observed within pingo cores along the southern shore of the proglacial river, as upheaved and slightly deformed strata (Allard, 1996).

The second half of the Holocene was marked by the deposition of glacio-fluvial, eolian and organic sediments over the valley floor. First, following marine regression after ~ 6 kyr BP, a glacial outwash plain probably occupied the entire valley, overlying the marine silts and clays and depositing glacio-fluvial sands and gravels (Allard, 1996, Fortier and Allard, 2004). Small streams with cold, alkaline, low-DOC and nutrient-poor waters were likely widespread within the plain, as inferred from diatoms observed at the base of the core collected in Gull Lake (Fig. 5; Antoniades et al., 2008; Zimmermann et al., 2010). The [14]C date of 4.8 kyr BP (5.5 cal. kyr BP) at the base of the core is interpreted as reworked pieces of peat transported by glacio-fluvial waters that sedimented in channels of the outwash (Table 1). This glacio-fluvial period lasted about two millennia, until eolian (fine sand and silt) and organic (peat) sediments started to accumulate in the valley around 3.7 kyr BP, as based on [14]C dating of *Salix* twigs and peat macrofossils (Fortier and Allard, 2004). A greater initial accumulation rate of > 2 mm yr$^{-1}$ occurred in this period, followed by a reduction to < 1 mm yr$^{-1}$ after 2.2 kyr BP (Allard, 1996; Fortier et al., 2006). Effective organic (peat) sedimentation during this period is further supported by the presence, in the Gull Lake core, of abundant benthic and epiphytic diatom species generally preferring moss substrates typical of more acidic peatland/wetland

environments (Antoniades et al., 2008; Pienitz, 2001; Zimmermann et al., 2010). These eolian and organic layers (stratified silt and peat) were gradually incorporated (syngenetically) into the permafrost as they accumulated and froze (see below). Syngenetic permafrost and associated ground-ice development accompanied this sediment deposition and surface aggradation in the Qarlikturvik valley, forming the polygonal terrace within which numerous ponds and lakes later formed and are still visible today (Fig. 1). During the late Holocene (roughly 3500 years ago), the valley floor had completely emerged from the

sea and the proglacial river running through the valley had started to cut into its own alluvial deposits. At the same time, cooler regional temperatures (the Neoglacial) resulted in slower melting of upstream glaciers, thus lower flow of the river, which enhanced the above-mentioned covering of the outwash plain by eolian and organic sediments (Fortier and Allard, 2004). Neoglacial cooling was reported at numerous sites across the eastern Canadian Arctic, based on diverse paleoenvironmental indicators (summarized in Fortier et al., 2006).

Repeated thermal frost cracking during severe winters had likely started as soon as the downstream portion of the valley was exposed (i.e. around 6000 yr BP) and later affected the whole glacio-fluvial outwash plain (and the upper section of the underlying marine clay unit), resulting in the formation of a first generation of ice wedges and related polygon networks. The development of the silty peat terrace, starting at 3000-3500 yr BP, caused a change in thermal contraction properties of the ground, triggering the formation of a second generation of ice-wedge polygons (Fortier and Allard, 2004). As a result, ice

wedges, several metres wide and 6-8 m deep, extend today through the sedimentary sequence, likely down to the upper section of the marine clays. These ice wedges define a complex patchwork of high-center and low-center polygons with diameters ranging from 5 to 40 m, with the top of ice-wedges coinciding with the base of the active layer (Allard, 1996). About two thousand years ago, the Gull Lake inception site was located in such a typical tundra landscape.

## 5.2 Thermokarst lake evolution: a conceptual model for syngenetic permafrost in formerly glaciated terrains

Based on our findings about the geomorphology and paleolimnology of Gull Lake's basin, we developed a conceptual model of its formation and evolution during the late Holocene, including potential future trajectories. This four-stage model is presented in Fig. 6.

We summarized in the previous section the initial conditions before the inception of Gull Lake (Stage 0), during the first half of the Holocene. At the beginning of the late Holocene, the site was characterized by a network of syngenetic ice wedges

extending through frozen peat, eolian silt and glacio-fluvial sands, and likely into marine clays at depth (Fig. 6a). The silty peat unit on the terrace is about 1.5-2 m in thickness with a volumetric ice content exceeding 50 % (Fortier and Allard 2004). Thawing of this unit under Gull Lake resulted in a ~ 0.7 m layer of thawed silty peat at the bottom of the lake (Fig. 4). The underlying glacio-fluvial unit is ice-poor and thus has a little subsidence potential upon thaw (Fortier and Allard, 2004). We explain the elevation gap between the maximum lake depth (~ 4 m) and the potential thaw subsidence of thermokarst-

susceptible sediments by the presence of a pre-existing depression 1-2 m deeper than the surrounding polygonal network. This depression was interpreted as a channel in the glacio-fluvial outwash underlying the silty peat, similar to channels observed today in the glacio-fluvial outwash in glacial valleys of Bylot Island. This depression likely collected snow and snowmelt

waters, especially during years of higher precipitation and weaker winds (lower evaporation). Such conditions resulted in active layer deepening and thermokarst initiation with ice wedge melting and development of small and shallow ponds, either over the ice wedges or at their junctions (Grosse et al., 2013) (stage 1; Fig. 6b). This stage of the model is supported by several field studies reporting thermokarst initiation starting from the top of melting of ice wedges (e.g., Abolt et al., 2020; French, 2017; Mackay, 2000; Ward Jones et al., 2020), rather than from the center of ice-wedge polygons (where ground ice content is much lower; Kanevskyi et al. 2017). This also indicates that micro-topography can induce thermokarst initiation, with minimal influence from regional climate variations (Biskaborn et al., 2013). This is illustrated by Gull Lake inception (i.e. transition from terrestrial to aquatic sedimentation), which was [14]C dated at around 2100 yr BP (Table 1 and Fig. 4), corresponding to the Neoglacial cooling period, characterized by lower-than-average air temperatures and also intervals of wetter-than-average local conditions (Fortier et al., 2006). With thermokarst and ponding, diatom communities changed from moss-associated or aerophilous (terrestrial) species to dominantly benthic or planktonic (aquatic) taxa typical of tundra ponds (Fig. 5) (Ellis et al., 2008). This mixed tundra landscape, combining terrestrial and freshwater environments, characterizes high-centered polygon networks that are observable today in the valley and elsewhere across the Arctic (e.g., Abolt, 2020; Kanevskiy et al., 2014; Ward Jones et al., 2020).

During autumn, heat loss from these small water bodies to the atmosphere and subsequent phase change of water to ice delayed the freezing front propagation in the underlying ground during the following winter (Kokelj and Jorgenson, 2013). Hence, this resulted in more efficient and deeper thaw during the following summer. Ponds then started to coalesce over and at the edge of ice-wedge polygons, extending the aquatic surface area over the terrestrial one (stage 2; Fig. 6c) (Shur et al., 2019). At this stage, however, the aquatic conditions likely did not last year-round in these water bodies, as they were shallower than the > 2-m thick ice cover that forms in winter. Shallow coalescent ponds that freeze to the bottom each winter are still a common feature in the valley today.

Eventually, lateral expansion by both thermal and mechanical erosion, as well as thaw consolidation and subsidence of the silts and peats beneath waterbodies, led to the formation of a lake *sensu stricto*, with unfrozen water beneath the ice cover throughout the winter (stage 3; Fig. 6d). Year-round aquatic conditions started to prevail, leading to the gradual accumulation of organic-rich lacustrine mud (gyttja). This is illustrated by the presence of abundant benthic and planktonic diatom species in the upper part of the analyzed core (0-10 cm; Figs. 4 and 5), typical of lacustrine ecosystems across several regions (Guiry and Guiry, 2019). However, the specific morphology of the lake, with a central deeper basin presenting relatively steep slopes surrounded by a gently sloping shallow platform, suggests that this evolution did not follow a strictly linear trend. The central basin, covered by at least 10 cm of lacustrine sediments at the coring site, appears notably older than the peripheral platform, where no or negligible gyttja was observed at 1-m depth (Video Supplement VS1). Assuming a relatively low sedimentation rate of 0.1 to 0.2 mm per year, typical of thermokarst lakes (Bouchard et al. 2011; Coulombe et al., 2016), the central basin was likely formed several centuries before the surrounding shallow platform.

Gull Lake's bathymetry clearly indicates that the lake bottom in the central basin is generally deeper than the maximum ice cover thickness. Hence, thermal conditions are in place for the development of a talik underneath the lake. The temperature

sensor installed within surface sediments of the deepest portion of the central basin in 2014-2015 showed that it never froze, although it stayed close to 1-2 °C for 9 months of the year, i.e. from mid-October to mid-June (Supplement S3). However, an unfrozen zone could not be clearly detected along the GPR lines (Fig. 3). Moreover, the bathymetry in the central basin is notably heterogeneous, with shallow parts (~ 2 m deep) in contact with winter ice cover (GPR line12; Fig. 3), and a deeper portion limited to the SW section of the lake (Fig. 2). The inferred talik is therefore not the typical bowl-shape that underlies the whole central basin, as generally reported from other thermokarst lake basins (Morgenstern et al., 2011 and references therein), but is closer to the configuration described by Burn (2002) for tundra lakes in the western Canadian Arctic, which have a talik under the deep central basin that is encircled by a shallow littoral terrace underlain by permafrost. A potential explanation for this heterogeneity in the deep basin bathymetry is the differential surface subsidence above ice wedges *vs*. above polygon centers, the latter being tempered by the presence of fibrous peat. Based on GPR lines, the bumpy lake bottom topography can be interpreted as the former surface of ice-wedge polygon ridges and troughs. Notably, numerous reflectors were identified along GPR lines underneath the lake bottom (at > 35 cm depth on average), and these reflectors were located 5 to 40 m apart laterally, similar to distances presently observed between ice wedges within the valley (Fig. 3) (Fortier and Allard, 2004; Fortier et al., 2019). The other (deeper) series of reflectors, found only underneath the peripheral platform (average depth of 2.6 m) in frozen material, likely represent the top of the glacio-fluvial unit (sand and gravel). This unit was indeed observed in the sediment core, which was collected in the deep basin in a non-frozen state (Fig. 4). Fortier and Allard (2004) reported a similar depth for the glacio-fluvial unit, based on permafrost coring and GPR surveys on the polygonal terrace a few tens of meters from Gull Lake's northern shore.

Gull Lake is currently slowly expanding laterally by thermokarst in the syngenetic frozen silt-peat terrace, and the 'thawing front' (i.e. the base of the talik) has now reached the underlying glacio-fluvial sand (Fig. 4; Fig. 6d). The future evolution of the lake towards its final disappearance might thus include one or both of the following scenarios: 1) a gradual terrestrialization through gyttja accumulation and lake infilling (stage 4a; Fig. 6e), 2) a rapid lateral drainage *via* shoreline breaching resulting from thermal erosion (e.g., thermo-erosion gullying, fluvial erosion) (stage 4b; Fig. 6f).

Lake or pond infilling causing terrestrialization has been reported from Arctic coastal Alaska (Jorgenson and Shur, 2007), subarctic eastern Canada (Payette et al., 2004), and boreal interior Alaska (Kanevskiy et al., 2014; Roach et al., 2011). We did not observe direct signs of terrestrialization at our study site. The absence of a visible inlet, as well as the relatively low concentrations of organic matter and nutrients measured at different times of the year (Bouchard et al., 2015a), might explain the slow sedimentation rate in Gull Lake. Added to current observations that 1) lake shores migrate laterally by both thermal and mechanical erosion, and 2) lake peripheral platform progressively deepens by thaw subsidence, it is likely that the complete lake infilling resulting in full-scale terrestrialization might not happen in the foreseeable future. Some partial infilling might have time to occur, but natural landscape evolution is likely to result in partial lake drainage, as suggested by the presence of numerous thermo-erosion gullies in the valley and by evidence of such a partial drainage in a nearby lake (Godin and Fortier, 2012) (Fig. 7).

Partial or complete drawdown of lakes can relate to their long-term water balance in relation to regional climate (e.g., precipitation *vs*. evaporation) (Bouchard et al., 2013a; Riordan et al., 2006). However, thermokarst lakes can also drain catastrophically, as has been reported from several permafrost regions across the Arctic (summarized for instance in Grosse et al., 2013 or Kokelj and Jorgenson, 2013). The drivers for such abrupt drainage are mostly related to local geomorphology and natural landscape evolution, hence factors that are external to the lakes themselves. These include ice wedge melting in the

surrounding basin creating a drainage network, retrogressive development of thermo-erosion gullies towards a given lake, coastal erosion or tapping by another lake or a river (French, 2017; Mackay, 1997). In the case of the Qarlikturvik valley in general, and Gull Lake in particular, it is likely that future evolution will involve lake drainage to a certain extent. First, networks of rapidly evolving thermo-erosion gullies, developed along melting ice-wedge networks, exist elsewhere in the valley (Fortier et al., 2007). Once initiated, these gully networks developed extremely rapidly during the first year (average

erosion rates of several meters per day) and at a much slower but quasi-steady rate afterwards for the following decade (Godin and Fortier, 2012). Such processes have strong impacts on local hydrology, including snow redistribution and surface/subsurface hydrological connectivity (Godin et al., 2014). Second, field observations suggest that a nearby lake located immediately downslope of Gull Lake (informally named "Gull Lake 2" or GL-2; Fig. 1c), has partly drained in the past. This is based on the observation of former lake shores, pingo development, and the absence of well-developed ice-wedge polygons

in the immediate surroundings of the remnant lake (Fig. 7). Such partial drainage is likely to happen to Gull Lake in the future, affecting at least the shallow peripheral platform and leaving a residual smaller lake corresponding to the current deeper basin.

**5.3 Implications for Arctic carbon dynamics**

Several square kilometres of the Qarlikturvik valley are currently underlain by a syngenetic permafrost terrace composed of alternating mineral (silt) and organic (peat) layers with an average organic matter content of 40 % (ranging from 15 to 65 %)

(Fortier and Allard, 2004). Assuming that bulk organic matter contains 58 % of organic carbon (Pribyl, 2010), the terrace contains more than 20 % of total organic carbon (TOC). This value is roughly one order of magnitude higher than the 2-3 % TOC values generally reported from the Yedoma domain of Siberia, Alaska and NW Canada (e.g., Schirrmeister et al., 2011; Strauss et al., 2017), which can be considered as a geomorphological analog. For this, we are assuming volumetric ground-ice content (40-70 %) and bulk density ($1-1.5 \times 10^3$ kg m$^{-3}$) that are comparable to other circumpolar regions (Fortier et al., 2006).

Since the thickness of the organic-rich permafrost terrace on Bylot Island (3-5 m) is roughly one order of magnitude lower than the average thickness of Yedoma deposits (30-50 m), specific carbon inventories in both types of landscapes can be considered more or less equivalent. However, this comparison does not take into account the lability of the organic matter, with much older parent material of a different diagenetic state in Yedoma landscapes (Mann et al., 2015; Vonk et al., 2013). Moreover, the comparison does not include any spatial considerations about the total carbon stocks: Yedoma landscapes are

estimated to cover ~ $1.0-1.4 \times 10^6$ km$^2$ (Strauss et al., 2017), whereas such numbers for syngenetic glaciated terrains are, to our knowledge, currently not available. There will likely be considerable regional differences in the biogeochemical response

to permafrost thaw, as thaw, and the associated carbon mobilization, are influenced by local relief and parent material (geology) (Tank et al., 2020; Williams and Smith, 1989).

Nevertheless, if future climate change results in even more widespread thaw of ice-rich permafrost (Biskaborn et al., 2019), it is plausible that the thawing of one meter of organic-rich frozen ground in this valley of Bylot Island could mobilize an order of magnitude more organic carbon ($\sim 200$ kg C m$^{-3}$) than an equivalent layer in Yedoma landscapes (20-30 kg C m$^{-3}$; Schirrmeister et al., 2011). These Bylot Island estimates are also much higher than those reported from the surface layer (0-1 m) of comparable ice-wedge polygon terrains developed in Holocene fluvial terraces in Siberia ($\sim 30$ kg C m$^{-3}$; Zubrzycki et al., 2013) and elsewhere across the continuous permafrost zone in cryoturbated organic/mineral soils called 'turbels' (ranging from $\sim 32$ to 61 kg C m$^{-3}$; Tarnocai et al., 2009). Therefore, the short-term carbon feedback potential caused by GHG emissions from landscapes as presented in this study is likely much higher than from Yedoma regions and many other ice-wedge polygon sites across the Arctic. For the specific case of Gull Lake, as the base of the talik has now reached the organic-poor layer of glacio-fluvial sands, as shown by the $^{14}$C maximum age of methane corresponding to the maximum age of the permafrost terrace (Bouchard et al., 2015; Fortier and Allard, 2004), future emissions from the deep basin are likely to slow down, although lateral expansion will likely continue to promote emissions from the current peripheral platform. To propose reasonable estimates from syngenetic glaciated terrains at the global scale, we not only need to know its global extent, but also the thickness and lability of the organic layer in a range of locations.

## 6 Conclusions

Combining high-resolution lake mapping (sonar and GPR), geomorphological observations and paleolimnological reconstructions (litho- and biostratigraphy) from Gull Lake on Bylot Island, we developed a conceptual model of thermokarst lake inception and evolution (past, present and future) in a syngenetic glaciated permafrost landscape of the eastern Canadian Arctic during the Holocene. The model explains multiple steps of local landscape evolution from terrestrial to freshwater environment. Paleoenvironmental reconstructions of formerly glaciated syngenetic permafrost landscapes are currently underrepresented in the thermokarst lake literature, which is dominated by Yedoma deposits (Pleistocene-age ice-rich permafrost). Moreover, this model explains the early development of thermokarst due to local topographic effects that promote top-down melting of ice wedges resulting in thermokarst ponds over the degrading ice wedges. This is followed by the subsidence of polygon margins into the thaw ponds as the ice-rich sediments thaw and consolidate. Over time the thermokarst ponds coalesce to form a thermokarst lake.

Based on our results, we conclude that thermokarst development during the Neoglacial at our study site was not mainly driven by warmer air temperatures. We rather infer that thermokarst, in the context of the regional climate history, was mainly driven by natural landscape evolution, or the complex feedback between local topography, ground ice degradation, snow cover distribution and depth, and surface hydrology. The existence of a pre-existing depression (abandoned glacio-fluvial outwash channel), collecting snow and meltwater and therefore affecting ground surface temperatures, underscores the control of paleo-

topography on permafrost landscape evolution. Thermokarst lake development on Bylot Island during the Holocene appears as a self-enhancing process occurring within a mature landscape. This process, once initiated, proceeds regardless of variations in air temperature. This is illustrated by Gull Lake inception (around 2000 years ago), which initially occurred during the cooler Neoglacial climate period, underscoring the importance of precipitation and local snow distribution over temperature alone.

The valley surrounding Gull Lake is occupied by a syngenetic permafrost terrace which, although much thinner than a Yedoma ice complex (3-5 m *vs*. 30-50 m, respectively), contains an order of magnitude greater amount of stored organic carbon per unit depth. Consequently, if an equivalent thickness of ice-rich permafrost thaws across the Arctic, the short-term carbon feedback potential caused by GHG emissions from this syngenetic glaciated permafrost landscape could be ten times higher than from Yedoma soils.

**Data availability**

The following related datasets are available in the Nordicana D collection at Centre d'études nordiques (CEN – Centre for Northern Studies) (http://www.cen.ulaval.ca/nordicanad/). The complete citations of each dataset appear in the reference list of this manuscript.

- Fortier et al., 2019: Ground-penetrating radar (GPR) survey data for a thermokarst lake, Bylot Island, Nunavut, Canada. doi: 10.5885/45609CE-E3573955017A4904.
- Fortier and Bouchard, 2019a: Computed tomography (CT) scans of a lake sediment core, Bylot Island, Nunavut, Canada. doi: 10.5885/45612CE-AB27C20EB10D4509.
- Fortier and Bouchard, 2019b: Organic matter content and grain size distribution in a lake sediment core, Bylot Island, Nunavut, Canada. doi: 10.5885/45603CE-21852993EE434926.
- Pienitz et al., 2019: Fossil diatom abundance in a lake sediment core, Bylot Island, Nunavut. doi: 10.5885/45600CE-C0960664FE8F4038.
- Fortier et al., 2020: Radiocarbon (14C) dates in terrestrial and aquatic environments, Bylot Island, Nunavut. doi: 10.5885/45651CE-C6FD628F45E44578

**Video Supplement**

**VS1**. Underwater camera video of submerged degraded ice-wedge polygons at the bottom of the peripheral shallow platform of Gull Lake, Bylot Island, Nunavut, Canada (as in Bouchard et al., 2015b). Water depth is approximately 1 m. Footage was collected in July 2014. doi: 10.5446/43923. Accessible at https://doi.org/10.5446/43923.

**Supplement Materials**

**1.** Computed tomography (CT) scanning of a 109-cm long sediment core collected in June 2015 in Gull Lake, Bylot Island, Nunavut, Canada. Methods as in Calmels and Allard (2004). 1 table (Table S1), 2 figures (Figs. S1-S2).

**2**. Plates (photographs) of the most abundant fossil diatoms found in a 109-cm long sediment core collected in June 2015 in Gull Lake, Bylot Island, Nunavut, Canada. Methods as in Bouchard et al. (2013b). 1 figure (Fig. S3).

**3**. Lake-water temperatures recorded at the bottom of a thermokarst lake (2014-2015), Bylot Island, Nunavut, Canada. 1 figure (Fig. S4).

**Author contribution**

FB, DF, RP and IL designed the research goals and methods. FB, MP and VB conducted a first analysis of the data and produced the figures. DF and IL funded fieldwork sampling and analyses. FB prepared the manuscript with contributions from all co-authors.

**Competing interests**

The authors declare that they have no conflict of interest.

**Acknowledgements**

We are grateful to the team of Gilles Gauthier (Dept. of biology, U. Laval), the Center for Northern Studies (CEN) and the staff of the Sirmilik National Park (Parks Canada) for logistical support and access to Bylot Island. We sincerely thank Vilmantas Prèskienis and Audrey Veillette for their leadership in conducting field surveys in 2015 (GPR and sediment coring) and for providing much needed help during an emergency. We also thank Jean-Philippe Tremblay and Yoan LeChasseur for
fieldwork preparation assistance, Maxime Tremblay for his help in the field, Arianne Lafontaine and Andréanne Lemay for their help in the laboratory, Louise Marcoux and Sylvie St-Jacques for their assistance in drafting figures, and Stéphanie Coulombe for inspiring discussions while preparing the first draft of the manuscript. We are also very thankful to the community and the people of Mittimatalik (Pond Inlet) for access to the territory. Finally, we thank two anonymous reviewers and the Editor for insightful and useful comments that greatly enhanced the manuscript. This research was funded by ArcticNet,
the Natural Sciences and Engineering Research Council of Canada (NSERC), the Polar Continental Shelf Program (PCSP) of Natural Resources Canada, the NSERC Discovery Frontiers grant "Arctic Development and Adaptation to Permafrost in Transition" (ADAPT), the EnviroNorth Training Program, and the W. Garfield Weston Foundation.

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

 **Tables**

Table 1: Radiocarbon ($^{14}$C) dates obtained in sediment cores collected in Gull Lake and in soil samples collected in the surrounding frozen silt-peat terrace [1].

| Sample ID | Environment | Dated material | Depth (cm) | 14C age (yr BP) | ± | Calib. age (cal yr BP) | 1-$\sigma$ range | Source |
|-----------|-------------|----------------|------------|-----------------|---|------------------------|------------------|--------|
| ULA-5673 | aquatic | lake sed. (bulk) | 9.25 | **2025** | 20 | **1972** | 1949 - 1995 | |
| ULA-5672 | aquatic | lake sed. (bulk) | 10.25 | **2100** | 20 | **2073** | 2042 - 2117 | |
| ULA-5959 | aquatic | wood/plant | 107.5 | **4805** | 15 | **5505** | 5488 - 5588 | Bouchard et al. |
| ULA-4894 | aquatic | lake sed. (bulk) | 0.5 | **1650** | 20 | **1552** | 1533 - 1563 | (this study) |
| ULA-4895 | aquatic | lake sed. (bulk) | 9.5 | **2065** | 20 | **2032** | 1993 - 2057 | |
| ULA-4896 | aquatic | peat | 53.5 | **2635** | 20 | **2756** | 2748 - 2760 | |
| Beta-143333 | terrestrial | peat | 297 | **3100** | 50 | **3303** | 3245 - 3374 | Ellis and |
| Beta-143337 | terrestrial | peat | 229 | **2590** | 50 | **2725** | 2541 - 2772 | Rochefort (2006) |
| Beta 143339 | terrestrial | peat | 209 | **1660** | 40 | **1565** | 1528 - 1611 | Ellis and |
| Beta 152437 | terrestrial | peat | 182 | **1470** | 40 | **1358** | 1316 - 1385 | Rochefort (2004) |
| UL-2356 | terrestrial | wood + peat | 233 | **3670** | 110 | **4010** | 3848 - 4151 | Fortier et al. |
| UL-2152 | terrestrial | wood + peat | 241 | **3270** | 100 | **3506** | 3387 - 3607 | (2006) |
| UL-1048 | terrestrial | peat | 135 | **2210** | 120 | **2208** | 2060 - 2346 | |
| UL-1034 | terrestrial | peat | 230 | **2510** | 90 | **2575** | 2489 - 2739 | Allard (1996) |
| UL-1035 | terrestrial | peat | 250 | **2600** | 90 | **2687** | 2496 - 2840 | |
| UL-1025 | terrestrial | peat | 320 | **2900** | 90 | **3045** | 2894 - 3165 | |
| ULA-6508 | terrestrial | peat | 82 | **3045** | 15 | **3249** | 3214 - 3323 | |
| UL-2427 | terrestrial | peat | 301 | **3040** | 90 | **3228** | 3080 - 3362 | |
| UL-2614 | terrestrial | peat | 275 | **3350** | 90 | **3593** | 3475 - 3693 | |
| UL-2418 | terrestrial | wood | N/A | **3560** | 90 | **3855** | 3720 - 3972 | Bouchard et al. |
| UL-2584 | terrestrial | wood | N/A | **3300** | 100 | **3537** | 3403 - 3640 | (this study) |
| UL-2416 | terrestrial | wood + peat | 155 | **3440** | 100 | **3706** | 3586 - 3832 | |
| UL-2264 | terrestrial | peat | 210 | **2750** | 90 | **2869** | 2765 - 2943 | |

[1] Other $^{14}$C dates obtained near the base of the frozen silt-peat terrace in the surroundings (see text for details) are also

 summarized (Allard, 1996; Fortier et al., 2006; Ellis and Rochefort, 2004; 2006). Complete unpublished data (Fortier et al., 2020) are available in open access files (see Data availability section).

**Figures**

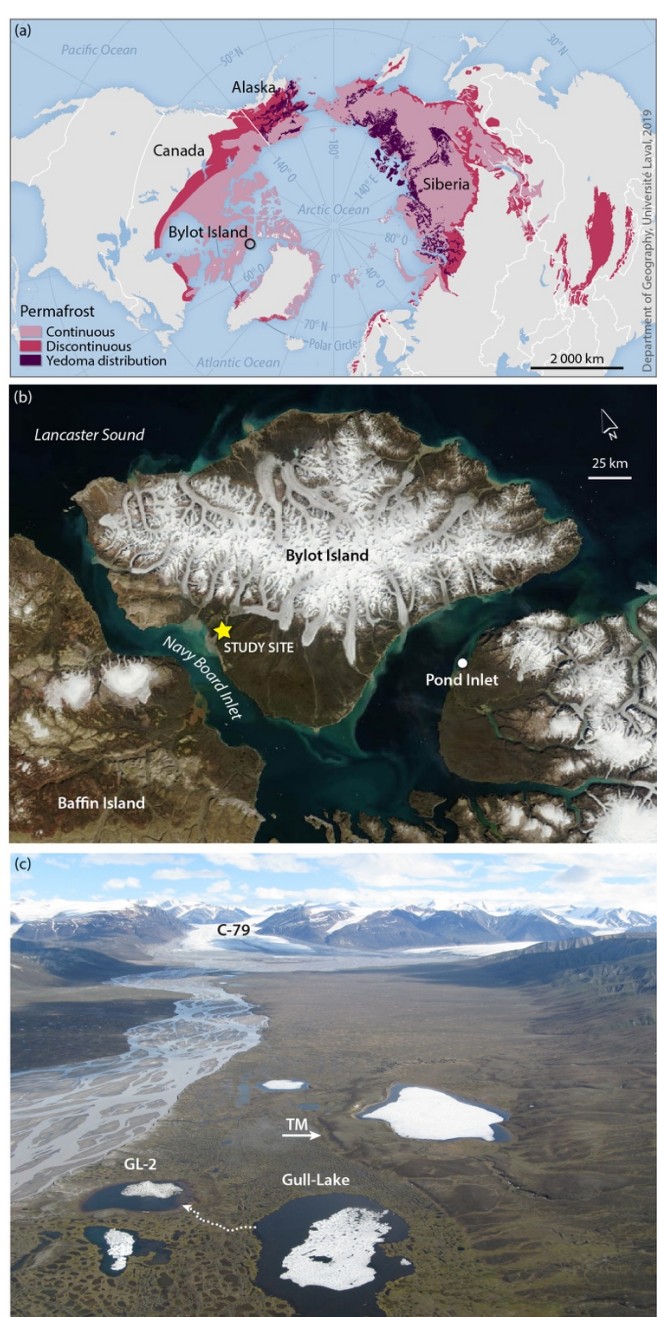

Figure 1: Study area location and context. a) Location of Bylot Island (Nunavut), Canada, within the continuous permafrost zone (source: Brown et al., 1998). Pleistocene ice-rich permafrost distribution in non-glaciated regions of Siberia and Alaska (Yedoma) is also shown (source: Strauss et al., 2017). b) Location of the study site, on the southwestern lowlands of Bylot Island (satellite photo: Terra-MODIS, 22 July 2012). c) Location of Gull Lake, in Qarlikturvik valley (glacier C-79 in the background). An early Holocene terminal moraine (TM) and a small outlet, draining towards "Gull Lake 2" (GL-2) and the proglacial river, are also shown.

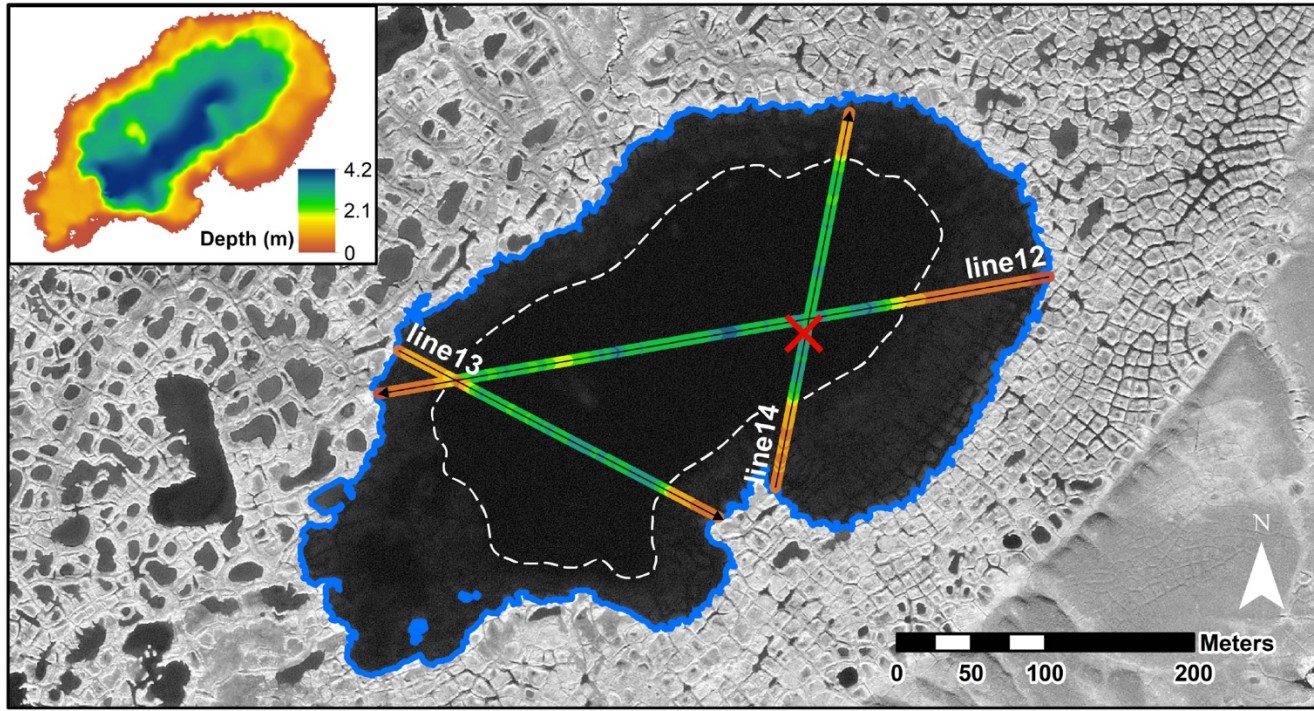

**Figure 2: Bathymetry and GPR survey lines conducted on Gull Lake. Sediment coring location is shown (red "x"). GPR line cross-sections (12, 13, 14) are shown in Fig. 3. The lake limit is delineated by a blue polygon. The central basin is deeper and surrounded by a shallow platform where degraded ice-wedge polygons are visible. The boundary between the central basin and the shallow platform is shown by the dashed white line. Satellite image: GeoEye-1, 18 July 2010.**

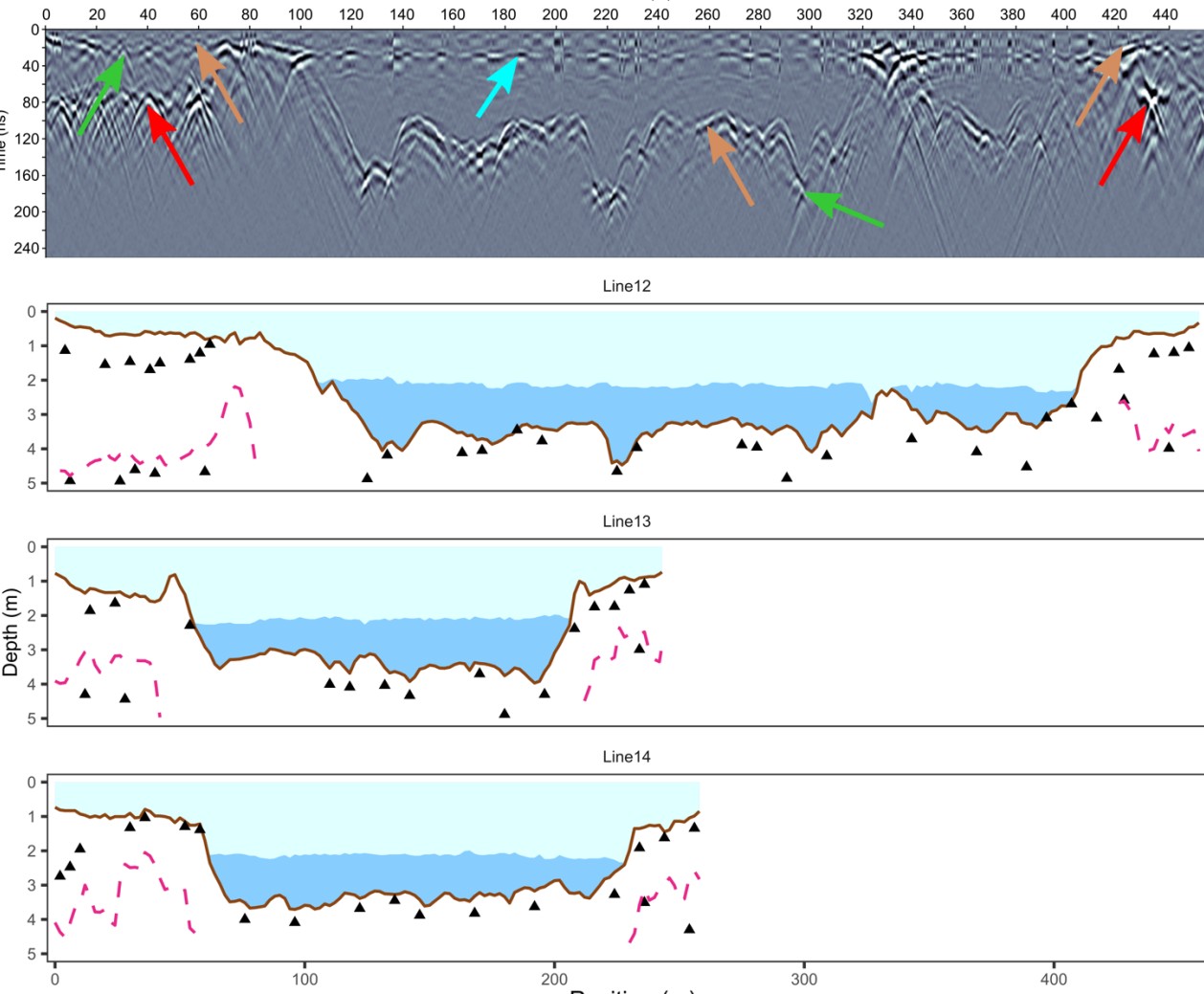

**Figure 3: Interpreted GPR cross-sections obtained along survey lines (see Fig. 2 for line locations). The upper figure is the raw GPR profile for line 12, with color arrows indicating distinct reflectors such as the base of the ice cover (light blue), lake bottom (brown), former surface of ice-wedge polygon ridges and troughs (green) and the top of the glacio-fluvial sand and gravel unit (red). Lower figures are interpretations, showing the ice cover (light blue area), free water area (dark blue area), lake bottom (brown line), glacio-fluvial stratigraphic contact (pink dashed line), and identified local reflectors, pictured as triangles. Complete data (Fortier et al., 2019) are available in open access files (see Data availability section).**

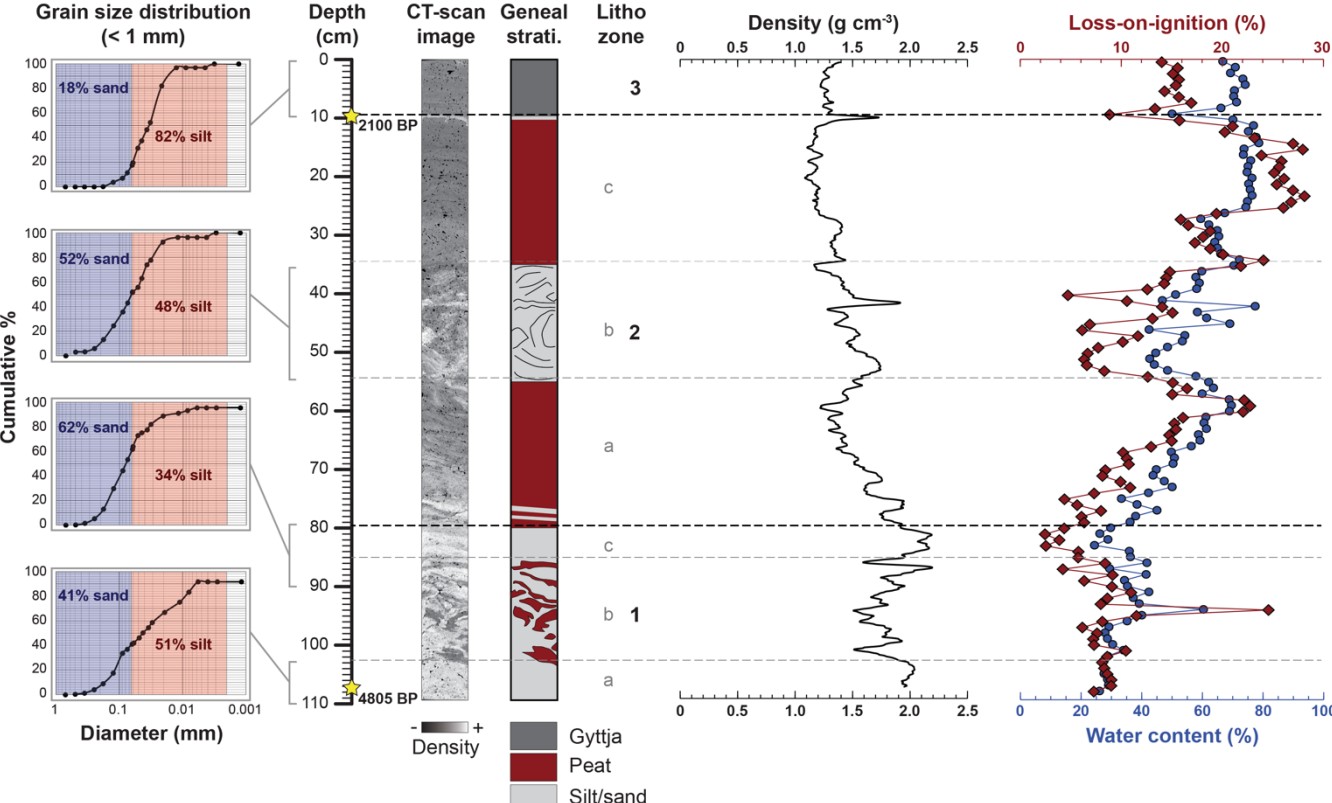

**Figure 4: Lithostratigraphy of a sediment core collected in Gull Lake in June 2015. The displayed CT-scan image, as well as visual descriptions and LOI data, were used to split the sedimentary sequence into 3 distinct units (lithozones). Complete data (Fortier and Bouchard, 2019a; 2019b) are available in open access files (see Data availability section). CT-scan details are summarized in Supplement S1.**

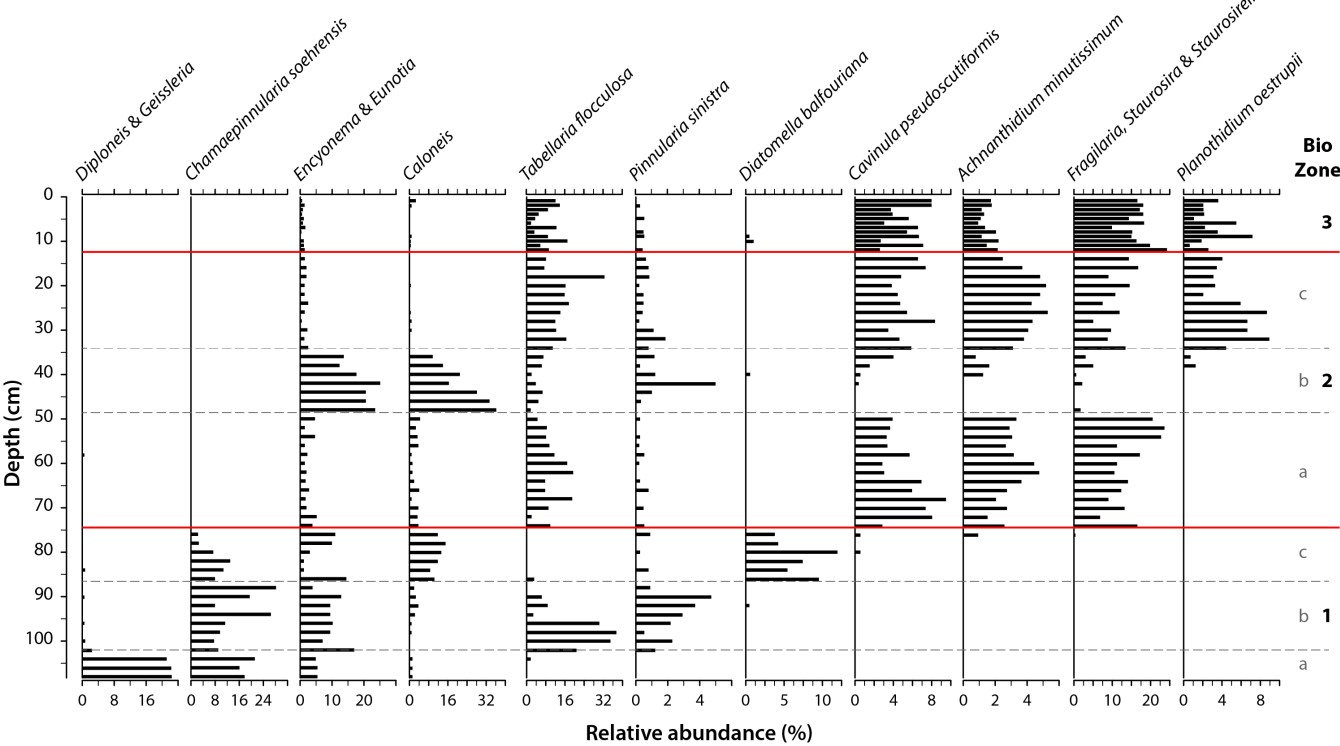

**Figure 5: Biostratigraphy (fossil diatoms) of a sediment core collected in Gull Lake in June 2015. Data are displayed as relative abundance (%) of dominant taxa, i.e. representing more than 5 % in at least one level. The relative abundance scale varies by taxa. Complete data (Pienitz et al., 2019) are available in open access files (see Data availability section). Plates (photographs) of the most abundant species are shown in Supplement S2.**

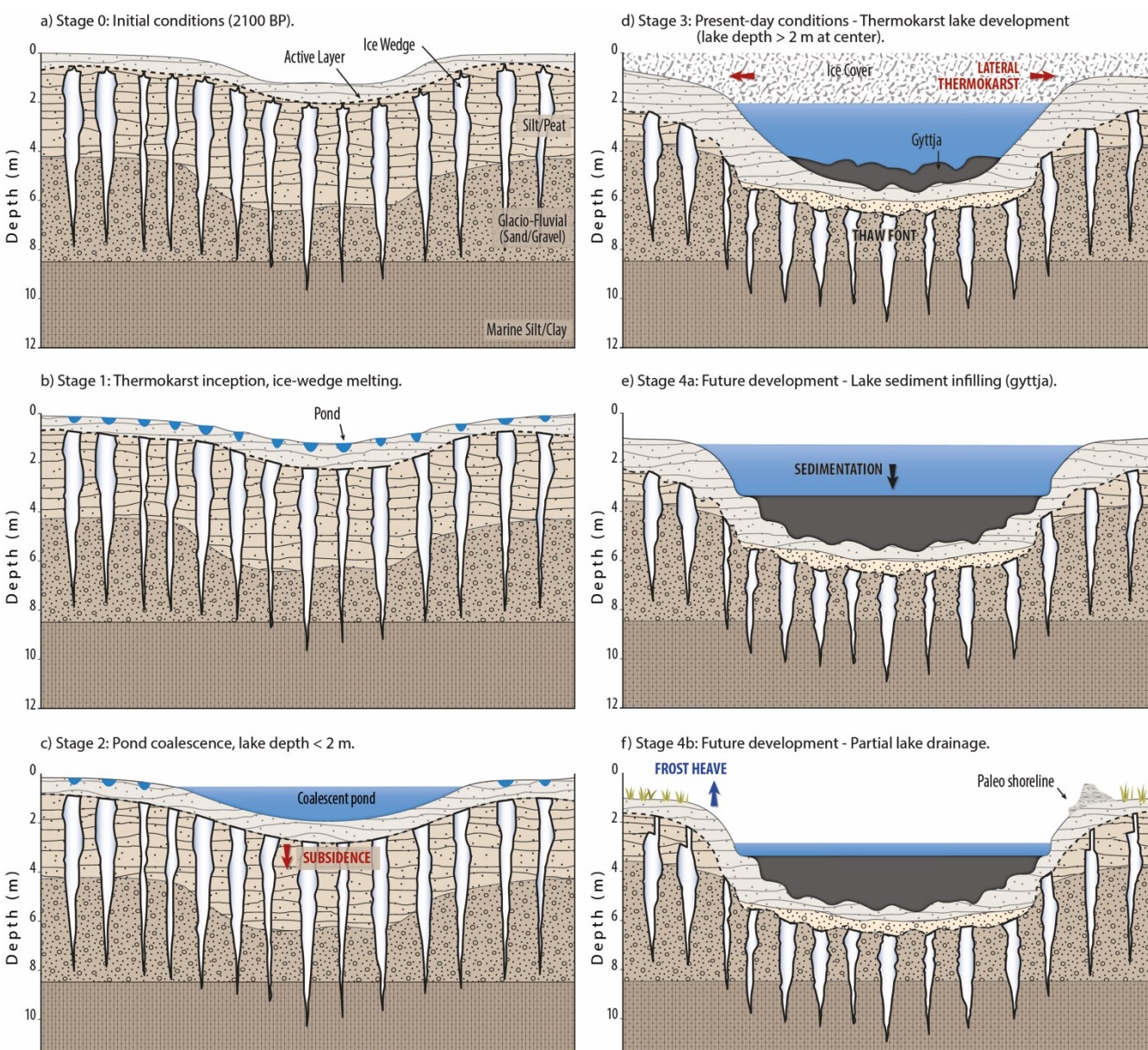

**Figure 6: Four-stage conceptual model of thermokarst lake inception and evolution through the Holocene. a) Stage 0: initial conditions with networks of ice wedges developed in frozen silt-peat and glacio-fluvial sand and gravel (and likely reaching underlying marine silts and clays). A pre-existing topographic depression of 1-2 m was collecting drifting snow and meltwater. b)**
**865 Stage 1: thermokarst inception, i.e. deepening of the active layer, melting of the top of ice wedges (triggering ice wedge truncation), development of a hummocky surface. c) Stage 2: thermokarst pond coalescence, formation of a small lake with a maximum depth still above maximum ice cover thickness. d) Stage 3: thermokarst lake mature development by lateral expansion (thermal and mechanical erosion) and bottom deepening (subsidence). Lake maximum depth is now below maximum ice cover thickness, triggering the formation of a talik. e) Stage 4a: possible future evolution by lake infilling (gyttja accumulation). f) Stage 4b: possible**
**870 future evolution by lake drainage (partial or complete) and re-activation of ice wedge cracking and growth (i.e., no more truncation).**

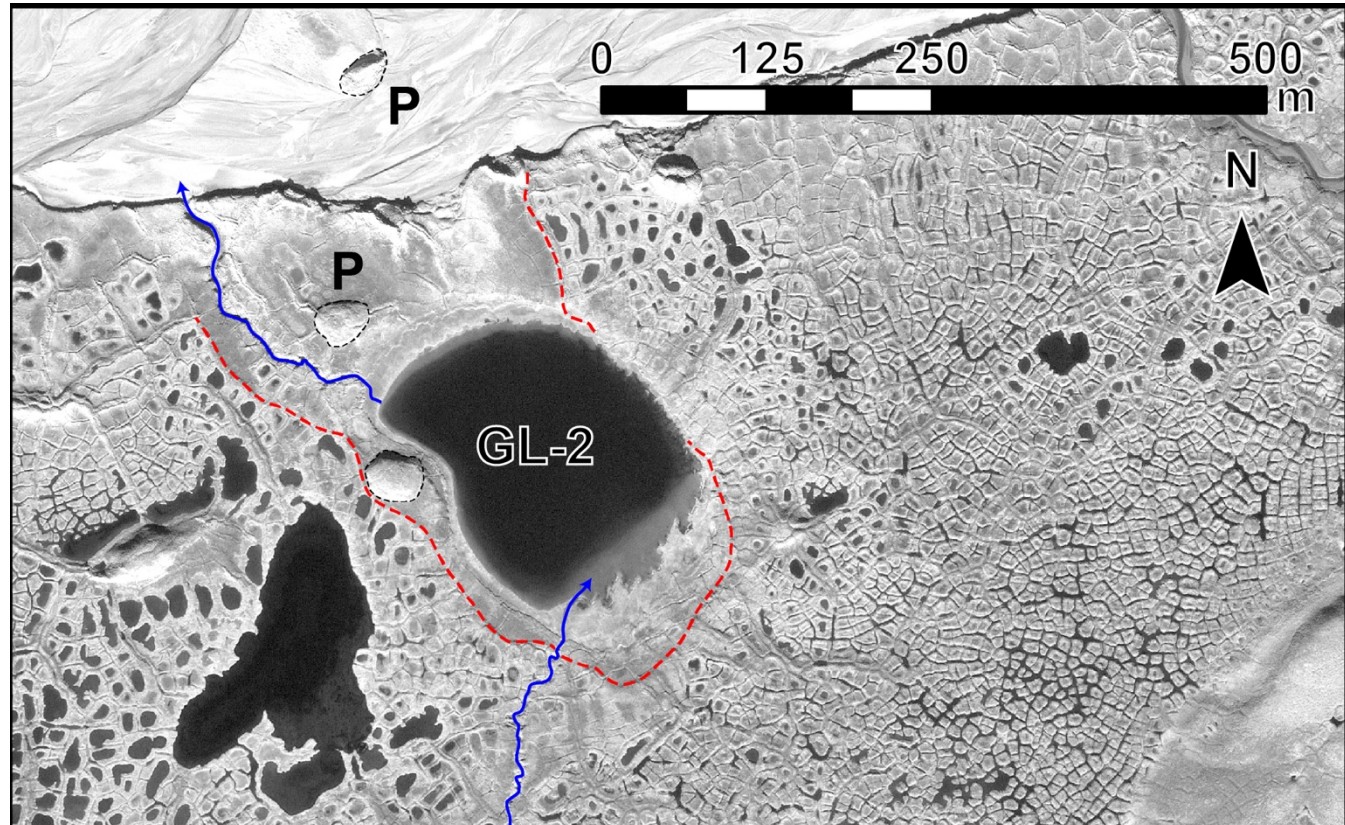

**Figure 7: Signs of past partial drainage around "Gull Lake 2" (GL-2). An inlet flowing from Gull Lake, located south, and an outlet draining towards the nearby proglacial river, are shown in blue. Former lake shores are shown by the dashed red lines. Pingos are indicated by a 'P'. Satellite image: GeoEye-1, 18 July 2010.**