# Peer review of "Thermokarst lake inception and development in syngenetic ice-wedge polygon terrain during a cooling climatic trend, Bylot Island (Nunavut), eastern Canadian Arctic"

_The Cryosphere, 2019_

## Referee Comment (RC1) · Anonymous Referee #1 · 10 Jan 2020

The authors present a case study in which they reconstruct the development of a thermokarst lake in Holocene-age sediments in the Canadian Arctic, primarily by interpreting lake geomorphology and a pair of sediment cores extracted from the bed. Overall the study is interesting, the topic is clearly appropriate for the journal, the data seem sufficient to support most of the conclusions, and the narrative is fairly clear. I also appreciated the video supplement, showing ice wedge furrows in the shallow platform of the lake. Two of the most interesting and novel aspects of the study are that the lake formed in recent sediments (instead of Pleistocene-aged Yedoma deposits) and its initiation was during a time that was cooler than the present – the authors address both these points, but I think they could do a better job of emphasizing them early on in

the paper, and even in the abstract. I am also a bit confused about part of the conceptual model—specifically, what happens to the sediments beneath the deepest part of the lake—as described in my last comment in the list below. My detailed comments, presented below, are listed by line number in the manuscript.

16: I think this is a good place to emphasize that the lake you studied is from the Holocene, as opposed to the Pleistocene. You could write "Here we present the gradual transition from syngenetic ice wedge polygon terrain to a thermokarst lake in Holocene sediments in the Eastern Canadian Arctic." Also, remove the s from "terrains."

26: I recommend emphasizing that the Neoglacial cooling period was cooler than today. For example, you could write ". . .this happened in the middle of the Neoglacial cooling period, likely under colder-than-present and wetter-than-average conditions."

37: Remove the word "a" in "a significant variability."

42: The sentence that begins with "Lakes located in. . ." is unclear. I think you are distinguishing between lakes that form in Yedoma and lakes that don't, but I'm not sure what you mean by the phrase "form a separate lake category." I recommend rewriting this sentence, emphasizing that 1) some thermokarst lakes that form in Yedoma can be up to several meters deep, and 2) the focus of this paper is on younger lakes that don't form in Yedoma.

54: Check to make sure the Cryosphere allows citations of papers in preparation. I'm not sure this is the case. This also applies to your citation of Tank et al. in line 415.

56-62: Are all thermokarst lakes inevitably destroyed by one of these mechanisms? You make it sound like this is the case.

86: Change "These glacial valleys. . ." to "The valleys of these glaciers. . ."

109-118: It would be nice to state the maximum depth of the lake somewhere in this paragraph.

147: Please define gyttja the first time you use this word.

175-176: Please specify what you mean by "plotted on diagrams." Is this describing Figure 5?

193: There is a Fortier et al. 2019 in your references, as well as a Fortier et al. 2019a and Fortier et al. 2019b. Please change this to a, b, and c and update your citations.

200: Change "hyperboles" to "hyperbolas."

280: Please provide more context for the sentence that begins "The fossiliferous marine sediments. . ." Right now it's difficult to figure out how it fits into the paragraph.

287: I think you mean 4.8 kyr and 5.5 kyr instead of 4.8 yr and 5.5 yr.

321-324: Please explain your reasoning more thoroughly in the sentence that begins "Based on present-day lake morphology. . ." It's difficult to figure out how you reached the conclusion that the initial depression in the surface must have been 1-2 m deep.

372-373: I'm confused by this part of the conceptual model. Please explain how "the deepest parts of the lake have now almost reached the underlying glacio-fluvial sand." (You also make this statement in lines 427-428.) As ground ice melts and subsidence occurs, the upper sediment layers reduce in thickness, but they typically are not removed. Are you indicating that the upper sediment layers beneath the lake bed are being removed as the lake expands, exposing the glacio-fluvial sands? If so, how is it that you can still see evidence of ice wedge polygon ridges and troughs in the deepest part of the lake bed (lines 363-367)?

---

## Referee Comment (RC2) · Anonymous Referee #2 · 10 Jan 2020

The manuscript "Thermokarst lake development in syngenetic ice-wedge polygon terrain in the Eastern Canadian Arctic (Bylot Island, Nunavut)" presents a careful study of thermokarst lake initiation outside of its main distribution area. Both sedimentation regime and ecology are reconstructed in this highly relevant study from the Canadian high Arctic, a region that is still vastly understudied, mainly due to its remoteness and challenging accessibility. The manuscript is very well written and well structured and presents its findings in a clear and concise way. The study works without an age depth model, but that cannot always be forced, especially in Arctic thermokarst lakes. The way the authors deal with this issue may be the most honest way to present the radiocarbon dates. The dates still give a general indication of the ages of the strata.

The authors present a new conceptual model of late Holocene thermokarst lake development. This landscape type and region is indeed strongly underrepresented in the thermokarst literature. I have listed a few general comments, detailed comments and minor edits below and advise to accept this manuscript after minor revisions.

General comments

1. The discussion should focus on/refer to the actual results more obviously. Large parts of the discussion are quite general.

2. I generally like that there is a section in the discussion dedicated to the wider implications of your findings. This section does, however, need some work still. My main concern here is that you are comparing syngenetic permafrost with Yedoma (which is also syngenetic permafrost). I am not convinced that the difference lies primarily in syngenetic vs. epigenetic permafrost. It is more a question of wetlands vs. non-wetlands. Formerly glaciated terrain often develops into wetlands studded with lakes, but there are also regions which were never glaciated and are rich in lakes and wetlands, e.g. the Arctic coastal plain of Alaskan or its continuation into Canada, or generally Beringian coastal lowlands. To me, the main difference lies in minerogenic vs. organic/peat deposits. Also, I am not sure the entire terrace you are studying is homogeneous in its organic matter content. Fortier and Allard, 2004, covered two low-centred ice-wedge polygons from the terrace, in which high organic matter contents can be expected, as these are usually wetlands. Your study looks at oone particular thermokarst lake. The findings are relevant, and it is also important to place the finding in a wider context, I am just not too happy with the emphasis on the quantitative comparison.

3. It should be made clearer what is new about the conceptual model. This can be done by editing the text only.

Detailed comments

Line 33: "remarkably" sounds a bit weird in this context. Please also reconsider the

phrase "circumpolar regions", as thermokarst lakes are strictly speaking most abundant in terrestrial Arctic lowlands.

Lines 42-44: Unglaciated ice-rich terrain is not necessarily Yedoma, it includes ice-wedge polygon peatlands and lowland thermokarst. Also, if you are categorizing lakes, you might have to be more explicit. Lakes in Yedoma terrain might still be thermokarst lakes, even if they tend to have a different morphology. Not sure this categorization is needed here.

Lines 50-51: "When thaw depth exceeds the maximum thickness of winter ice cover, [...]" - this is ambiguous. Please rephrase.

Line 60: "drawdown" - this might not be the most appropriate term here, especially when you are also using it for lake infilling, could use "lowering" or "decreasing lake depth" instead

Lines 82-83: your third objective could end with "specifically for syngenetic ice-wedge polygon terrain" or else convince me that your conceptual model is universal

Line 99: Did glacier retreat stop for good or is it retreating again now? Also, please give a reference for the date.

Line 106: not sure "off-shore" is the appropriate term here, it sound like way off the sea shore

Line 122: Consider indicating that this publication describes the method. It sounds like the results have already been published.

Lines 175-176: could refer to the full diatom data set in the database

Lines 192-193: Consider indicating that this is data, not a published study. It sounds like the results have already been published.

Line 228: Do you trust the age at this depth? It sounds a bit old for a depth of 10cm, and as you dated bulk sediment, how can you be sure this is not, at least in part,

relocated old material? Please comment shortly.

Line 233: Is "major taxa" a known term? If the 5% abundance in at least one sample (?) is a commonly used standard cutoff, please give a reference. I know this is done to minimize statistical issues arising from extremely low counts and too many zero entries ("not present") in the dataset, but you could consider stating this in the text.

Line 234: "one level" – please specify what you mean here, one zone, one sample or something else?

Line 315: You could start this section with "Based on our findings on the geomorphology and palaeolimnology of..." to make it clear that you are talking about the new results rather than about findings from the literature. Or if it is both this study's findings and the results of your lit review, you can say so. This would make it clearer what you have added to the knowledge on thermokarst lake evolution in the region. You can also use active voice in the statement following in line 318 ("we summarized the initial conditions ..." or something similar).

Line 323: "must have" – I am not too happy with this absolute phrasing. Please also prove this and give references

Line 336: Give reference. Also, consider changing "high-centered polygons" to "ice-wedge polygons" in general, all types of which provide a mosaic of terrestrial and freshwater ecosystems in very close proximity of each other:

e.g. Bliss L.C. 1956. A Comparison of Plant Development in Microenvironments of Arctic and Alpine Tundras. Ecological Monographs 26, 303-337, 10.2307/1948544.

something newer:

from Siberia: De Klerk P., Teltewskoi A., Theuerkauf M. & Joosten H. 2014. Vegetation patterns, pollen deposition and distribution of non-pollen palynomorphs in an ice-wedge polygon near Kytalyk (NE Siberiawith some remarks on Arctic pollen morphology. Polar Biology, 1393-1412, 10.1007/s00300-014-1529-3.

From western Canada: Wolter J., Lantuit H., Fritz M., Macias-Fauria M., Myers-Smith I. & Herzschuh U. 2016. Vegetation composition and shrub extent on the Yukon coast, Canada, are strongly linked to ice-wedge polygon degradation. 2016, 10.3402/polar.v35.27489.

This is also found in palaeostudies using biological proxies, including diatoms: Fritz M., Wolter J., Rudaya N., Palagushkina O., Nazarova L., Obu J., Rethemeyer J., Lantuit H. & Wetterich S. 2016. Holocene ice-wedge polygon development in northern Yukon permafrost peatlands, Canada. Quaternary Science Reviews, 10.1016/j.quascirev.2016.02.008.

Line 352: Is this really typical of thermokarst lakes? See if other references state other accumulation rates, perhaps check

Biskaborn, B.K., Herzschuh, U., Bolshiyanov, D. et al. J Paleolimnol (2013) 49: 155. https://doi.org/10.1007/s10933-012-9650-1,

Klein et al., 2013 - https://doi.org/10.1016/j.palaeo.2013.09.009 or similar.

There is slightly more data on carbon accumulation rates in thermokarst lakes (e.g. Anthony, K., Zimov, S., Grosse, G. et al. A shift of thermokarst lakes from carbon sources to sinks during the Holocene epoch. Nature 511, 452–456 (2014) doi:10.1038/nature13560)

Sediment accumulation in thermokarst lakes has been shown to be messy and not at all constant. It can also be fairly high. See for example :

Schleusner et al., 2015 doi:10.1111/bor.12084

or Lenz et al., 2016 https://doi.org/10.1007/s41063-016-0025-0

or Wolter et al., 2017 https://doi.org/10.1177/0959683617708441

Line 356-358: You are introducing new results here (technically). Perhaps mention this earlier in the manuscript in the appropriate sections (methods/results)? I haven't seen

it there. This is not major new data, though, so you might also keep it as it is. Also, the results section did not state clearly that an unfrozen zone could not be detected from GPR data.

Lines 372-376: Some questions from my side: Do you mean that the lake cannot become any deeper once it hits the glacio-fluvial sand because of its lower ice centent? And do you think it could also stay there without disappearing? Is it certain the lake would drain because of topography or could it also coalesce with the lake next to it? Do you think lake infilling is really an option, as accumulation rates are low and the lake might grow both laterally and vertically in the future?

Line 377: Infilling by aquatic and semiaquatic plants is, to my knowledge, more likely in smaller ice-wedge ponds than in lakes. I agree that basins usually fill up with sediments over time, but this might takes a very long time. Jorgenson and Shur, 2007, are talking about infilling ponds along the margins of drained lake basins (while large thaw lakes may form in their centers) on the Arctic Coastal Plain of Alaska. The question is the balance between accumulation and decomposition or transport out of the system (e.g. via a stream), or in this case, possibly also lake deepening through additional thaw subsidence. Lake infilling would likely be a very slow process, especially given the low sediment accumulation rates.

Line 379: This argument is not super-convincing. Lacustrine sediments normally accumulate in a lake, so that you found them does not necessarily prove terrestrialization.

Line 391: give reference(s)

405-410: You could state here for which depths the Yedoma TOC contents were calculated, so it is comparable to your findings. Generally, I think you should not extend the carbon contents of the upper 3-5 m to greater depths, as TOC generally decreases strongly below the first meter or so. Also, to compare a point measurement from organic-rich sediment in a relatively small feature in a heterogeneous landscape with averaged values for the entire Yedoma domain is a bit misleading.

Line 415: Please avoid citing articles in review, as they are not available for checking.

Lines 424-427: This could be 2 sentences. Be extra careful in your phrasing here: "the entire Yedoma complex" sounds like you mean all of the Yedoma there is (its entire area). Also, as commented above, be careful with that comparison. It is valid to say that there are landscapes in the arctic that contain more organic-rich sediments than found on average in Yedoma, but this quantifying comparison is going too far for my taste, see my general comment above. Consider citing GHG emissions from Arctic wetlands. And how do your findings relate to findings from other Arctic wetlands or other low-centered polygon fields? You could for example compare your findings to those from ice-wedge polygon wetlands in the western Canadian Arctic, i.e. on the Yukon Coastal Plain, the eastern part of which used to be glaciated, or on the Tuktoyaktuk Peninsula .

Line 439: Thermokarst lake cycles would take far longer to develop. There simply wasn't enough time for that. The study design was thus not suitable for testing whether thermokarst lakes develop cyclically or unidirectionally.

Lines 437-438: Not all Pleistocene-age permafrost deposits are Yedoma.

Lines 441-444: "regardless of climate" – that might be a bit too much. In your next sentence you rightly state that precipitation (which is also climate) and snow distribution (which has a geomorphological component) are more important than temperature development. Stick with that.

Technical edits

line 43: replace "since" with "in"

line 54 and elsewhere: is the "in prep." manuscript published now? If not, omit reference.

Line 81: omit "a" before "syngenetic ice-wedge polygon terrain"

Line 87: developed into

Line 91: total precipitations -> a total precipitation

Line 93: perhaps better to separate this insert with commas instead of brackets

line 105: "before 3700 years ago" - sometime before? Just before?

Line 128: not sure one can "conduct" survey lines?

Line 198: the lake bottom

Line 218: "..., which are both dominated by peat"?

Line 254: "in average" should be "on average"

Line 268: separate the insert however by commas (was, however, strongly)

Line 269: Better use "entire" instead of "whole"?

Line 297: Perhaps better to say "During" or "At the beginning of the" late Holocene

Line 323: "found at the lake bottom"

Line 337-338: "were a significant source of latent heat to extract in autumn" – this needs some rephrasing. I do not much like the use of the word significant outside when not talking about statistical significance. And I do not quite understand the word "extract" in this context.

Line 354: the lake bottom

Line 386: water balances -> water balance

Line 392: tapping

Line 408: "presents slightly over" could be "contains more than"

Line 411: "are comparable to other circumpolar regions"?

Line 418: formerly glaciated terrain ?

Line 442: "self-enhancing"?

Figure 5: Is it possible to add ecological interpretation/groups on top of the taxa? That might help readers.

Figure 6: Is this conceptual model really new? It looks a lot like the existing models. The only immediate difference I see are the ponds forming on top of the ice wedges instead of between ice-wedge ridges. This might be because of climatic warming and subsequent ice-wedge degradation. How can you prove that the ponds were on top of the ice wedges? Is there a difference in diatom flora between intrapolygonal and interpolygonal ponds?

---

## Author Response (AR1)

We thank the two anonymous Reviewers and the Editor for their reviews of our manuscript and their useful comments. Below are point-by-point responses to all of the comments and questions. The original comments are shown in *grey (italics, smaller font)*, and our responses are presented in black (normal font). Please note that **the line numbers mentioned in our responses refer to the marked-up version** ('track changes' mode), attached at the end of this letter (thus not the 'clean' revised manuscript).
* * *
**Editor (Peter Morse)**
Received: 11 February 2020

*Thank you for your careful consideration of the reviewer's concerns. It is clear from the comments that the reviewers thought quite favourably of your manuscript, and provided many helpful comments that I think will improve the final paper. I would like you to please submit a revised manuscript for me to review.*

Thank you. We have prepared a revised manuscript and we attached the marked-up version at the end of this letter.

*Reviewer 1 made many helpful and detailed comments. Most appear to be easily dealt with, and I have just a few follow up questions that I hope you can clarify at this stage.*

*Line 99: Please ensure there is a commensurate change in the text to reflect your response to the comment.*

We added a sentence in the text (lines 122-124), including a new reference, and we also updated our response to this comment (see below, p. 12).

*Line 228: Again, please ensure that there is a commensurate change in the text to reflect your response to the comment.*

Here as well, we added a sentence in the text (lines 277-278) and we also updated our response to this comment (see below, p. 13).

*Lines 356-358: If these are results. It may be that these data were not in the original study design, and yet they were not produced by accident. I would like you to please include the results in the results section and discuss here. Or, if the data are already published in Preskienis et al., then you may want to keep here as a part of the discussion of your results in the context of other work. There needs to be a clear line between your contribution and that of others in the extended research group. Somebody needs to take responsibility for the data and explain its origins.*

We now mention these data as new sentences into the methods and the results sections (lines 170-172; 247-249). This is in addition to what we had already included in the discussion (lines 443-445).

*Line: 415: Here and elsewhere, please see if you can replace such articles "in review" with materials that have been published. Everybody wants to be able to follow up on referenced statements, and this is not possible for materials in review (that may not get published). Please replace those "in preparation" with an alternative.*

As asked, we removed or replaced all the 'in preparation' and 'in review' manuscripts. When there was only one citation, we replaced it (example at line 87), and when there were several citations that already supported the statement we simply removed the 'in review' citation (example at lines 143-144).

*Lines 424-427: Please do include a few sentences that compare your site to other ice-wedge polygon sites.*

Among the papers reporting on greenhouse gas emissions from other, comparable ice-wedge polygon sites (e.g., Abnizova et al., 2012), we could not find any that mentioned the specific organic carbon stocks in soils (in kg C m$^{-3}$ for the first meter below surface, or in kg C m$^{-2}$ integrated over the whole soil profile). We could find one paper reporting on organic carbon stocks from ice-wedge polygon terrain in the Lena Delta (Zubrzycki et al., 2013). In a broader context, the synthesis from Tarnocai et al. (2009) reports on organic carbon stocks in permafrost across the circumpolar north, especially the so-called 'turbels' (organic/mineral frozen soils of the continuous permafrost zone, often cryo-turbated), typical of ice-wedge polygon tundra landscapes. For all these sites, reported values generally range from 1 to 125 kg C m$^{-3}$, with an average near 30 or 60 kg C m$^{-3}$ within the 0-1 m layer, depending on the method used (Tarnocai et al., 2009; Tables 4 and 5).

We thus added the following sentence:
« These Bylot Island numbers are also much higher than those reported from the surface layer (0-1 m) of comparable ice-wedge polygon terrains developed in Holocene fluvial terraces in Siberia (~ 30 kg C m$^{-3}$; Zubrzycki et al., 2013) and elsewhere across the continuous permafrost zone in cryoturbated organic/mineral soils called 'turbels' (ranging from ~ 32 to 61 kg C m$^{-3}$; Tarnocai et al., 2009). » (lines 534-537)

We also modified the next sentence:
« Therefore, the short-term carbon feedback potential caused by GHG emissions from the present study landscape is likely much higher than from Yedoma regions and many other ice-wedge polygon sites across the Arctic. » (lines 538-539)

*Figure 6: If the reviewer was not convinced as to the novelty of your conceptual model, you will need to think carefully about it. What is new, and what is in agreement with existing models. I think that you have an opportunity to point out that models that suggest that thermokarst initiates from the low polygon is not in agreement with what we see in the field. There is a lot of literature on thermokarst initiation at the ice wedges (e.g., Mackay 2000, Figure 3), and your model agrees with it.*

As answered to the reviewer (p. 22-23 below), we see at least two novelties in our model:
1) ground ice melting and thermokarst initiation starting from the top of ice wedges, (ice wedge junctions or along longitudinal segments), rather than at the center of ice wedge polygons;
2) thermokarst lake inception and development during a cool climate episode (Neoglacial), underscoring the role of precipitation (especially winter snow cover) over increasing temperatures only.

Regarding 1), we added the following sentences in the discussion and in the conclusions:
« Thermokarst initiation starting from the top of melting of ice wedges, rather than from the center of ice-wedge polygons (where ground-ice content is much lower), agrees with many previous studies (e.g., Abolt et al., 2020; French, 2017; MacKay, 2000; Ward Jones et al., 2020). » (lines 418-421)
« Moreover, this model explains the early development of thermokarst by the local topography and top-down melting of ice wedges, followed by the subsidence of polygon centers. » (lines 568-569).

Also: if the Editor finds it relevant/helpful, we could add one of the pictures attached to this report (p. 15 or p. 23) as supplement material to demonstrate our point about melting ice wedges more explicitly.

*Reviewer 2 also provided several helpful comments. The only outstanding issue is with respect to the first general comment. R2 would like you to be more pointed in your discussion. Keeping in mind that you have addressed the detailed comments, please review your discussion carefully and see if you can tease out a few more subtleties that draw specifically from your work. It is these non-general details that will help increase the impact of this work, and distinguish it from other research.*

The early discussion (5.1) is necessary to 'set the stage' for our model. We nevertheless included some of our actual results (e.g., lines 339-343; 347-349). Then, we also modified several sentences and added new ones in order to put more 'beef' in our discussion. These changes refer to, among others, the pre-existing topographic depression (lines 377-384), the importance of melting ice wedges delimiting high-centered polygons (lines 385-390; 418-421) and the comparison with other ice-wedge polygon terrains in the Arctic (lines 534-537; 555). Finally, we modified the conclusions accordingly (lines 567-569; 570-575).

**Reviewer #1 (Anonymous)**

Answers posted online: 06 February 2020

General comments

*The authors present a case study in which they reconstruct the development of a thermokarst lake in Holocene-age sediments in the Canadian Arctic, primarily by interpreting lake geomorphology and a pair of sediment cores extracted from the bed. Overall the study is interesting, the topic is clearly appropriate for the journal, the data seem sufficient to support most of the conclusions, and the narrative is fairly clear. I also appreciated the video supplement, showing ice wedge furrows in the shallow platform of the lake.*

Thank you for these positive comments. The video supplement clearly shows that the lake is currently developing by lateral expansion related to thermokarst processes.

*Two of the most interesting and novel aspects of the study are that the lake formed in recent sediments (instead of Pleistocene-aged Yedoma deposits) and its initiation was during a time that was cooler than the present – the authors address both these points, but I think they could do a better job of emphasizing them early on in the paper, and even in the abstract.*

We modified several sentences, and added new ones, to put more emphasis on these two aspects. The fact that lake initiation started in Holocene sediment during a colder climate has important implications for thermokarst modeling. The abstract, the Introduction, as well as the Conclusions, were modified accordingly. We also changed the title, which now reads as follows: « Thermokarst lake inception and development in syngenetic ice-wedge polygon terrain during a cooling climatic trend, Bylot Island (Nunavut), Eastern Canadian Arctic ».

*I am also a bit confused about part of the conceptual model […] specifically, what happens to the sediments beneath the deepest part of the lake […], as described in my last comment in the list below.*

See our reply to the last comment below. The confusion came from our inaccurate wording. We have changed the text.

*My detailed comments, presented below, are listed by line number in the manuscript.*

Thank you. We replied to all comments and changed the text accordingly.

Specific comments

*16: I think this is a good place to emphasize that the lake you studied is from the Holocene, as opposed to the Pleistocene. You could write "Here we present the gradual transition from syngenetic ice wedge polygon terrain to a thermokarst lake in Holocene sediments in the Eastern Canadian Arctic." Also, remove the s from "terrains."*

We made the suggested changes.

*26: I recommend emphasizing that the Neoglacial cooling period was cooler than today. For example, you could write "this happened in the middle of the Neoglacial cooling period, likely under colder-than-present and wetter-than-average conditions."*

We changed the sentence as suggested. As mentioned above, we also modified the title to take that comment into account.

*37: Remove the word "a" in "a significant variability."*

Removed.

*42: The sentence that begins with "Lakes located in […]" is unclear. I think you are distinguishing between lakes that form in Yedoma and lakes that don't, but I'm not sure what you mean by the phrase "form a separate lake category." I recommend rewriting this sentence, emphasizing that 1) some thermokarst lakes that form in Yedoma can be up to several meters deep, and 2) the focus of this paper is on younger lakes that don't form in Yedoma.*

We modified the beginning of the sentence as follows: «Some of the lakes located in […].». Also, we added a sentence afterwards to mention that most thermokarst lakes are located in formerly glaciated terrains (i.e. Yedoma lakes are an exception).

*54: Check to make sure the Cryosphere allows citations of papers in preparation. I'm not sure this is the case. This also applies to your citation of Tank et al. in line 415.*

We included the Preskienis et al. paper because we knew that it would be submitted quickly after our own submission. It has been submitted since then (November 2019) and is now 'in review'. We modified the citations in the text and the reference list accordingly. Regarding the Tank et al. paper, it was already 'in review' in late 2019, and it still is.

Message to the Editors: We hope that these papers will be accepted for publication by the time our manuscript is published. Meanwhile, we can provide a copy of these

manuscripts (read-only) for reference to the reviewers. If this is not OK with the Editors, we can find another solution.

*Update (21 February 2020):*
We removed all the 'in prep' and 'in review' citations.

*56-62: Are all thermokarst lakes inevitably destroyed by one of these mechanisms? You make it sound like this is the case.*

The outcomes presented are the ones we know about (they have been studied, and we refer to these studies). To be more cautious, we added 'generally': « […] thermokarst lake development generally ends with one or more of the following […] ».

*86: Change "These glacial valleys […]" to "The valleys of these glaciers[…]"*

Change made.

*109-118: It would be nice to state the maximum depth of the lake somewhere in this paragraph.*

Detailed lake morphology (including maximum depth) is provided in the Results section. However, we added the maximum depth in this paragraph, as suggested: « The sampled lake, informally named Gull Lake (maximum depth ~ 4.2 m), is located […] ».

*147: Please define gyttja the first time you use this word.*

Done: « […] general stratigraphic units, such as gyttja (organic-rich lacustrine mud), peat, silt and sand […] ».

*175-176: Please specify what you mean by "plotted on diagrams." Is this describing Figure 5?*

Yes, it is referring to Figure 5. We changed the sentence into: « […] were displayed on abundance diagrams using the C2 software […].

*193: There is a Fortier et al. 2019 in your references, as well as a Fortier et al. 2019a and Fortier et al. 2019b. Please change this to a, b, and c and update your citations.*

These references are all recent datasets involving either only two authors (Fortier and Bouchard, 2019a and 2019b), or more than two authors (Fortier et al., 2019). These references are as follows (we also specified this in the 'Data availability' section):

- Fortier, D., and Bouchard, F.: Computed tomography (CT) scanning of a lake sediment core, Bylot Island, Nunavut, Canada, v. 1.0 (2015-2015), Nordicana D54, doi: 10.5885/45612CE-AB27C20EB10D4509, 2019a.
- Fortier, D., and Bouchard, F.: Loss-on-ignition and grain size analysis of a lake sediment core, Bylot Island, Nunavut, Canada, v. 1.0 (2015-2015), Nordicana D52, doi: 10.5885/45603CE-21852993EE434926, 2019b.
- Fortier, D., Paquette, M., and Bouchard, F.: Ground-penetrating radar (GPR) surveys over a thermokarst lake, Bylot Island, Nunavut, Canada, v. 1.0 (2015-2015), Nordicana D53, doi: 10.5885/45609CE-E3573955017A4904, 2019.

*200: Change "hyperboles" to "hyperbolas."*

Done.

*280: Please provide more context for the sentence that begins "The fossiliferous marine sediments […]" Right now it's difficult to figure out how it fits into the paragraph.*

We are referring to the silts and clays deposited by the marine transgression phase in the sentence just before. We slightly modified the text, so now it is more explicit: « Such fossiliferous marine sediments […].

*287: I think you mean 4.8 kyr and 5.5 kyr instead of 4.8 yr and 5.5 yr.*

The reviewer is right, good catch on this mistake. We changed the text accordingly.

*321-324: Please explain your reasoning more thoroughly in the sentence that begins "Based on present-day lake morphology […]" It's difficult to figure out how you reached the conclusion that the initial depression in the surface must have been 1-2 m deep.*

The reasoning is as follows:
In the sediment core from 2015, collected at ~ 4 m depth, we sampled about 0.7 m of silty peat. This unit is currently unfrozen. We know that the surrounding frozen ground of that unit contains over 50 % of ice by volume (Fortier and Allard, 2004). Hence, considering thaw settlement and consolidation, the silty peat layers found in the core must have made at least twice their current thickness when they were still frozen. That makes about 1.5-2 m thick of frozen silty peat before the lake started to form. Even if we assume that the thawing of the underlying glaciofluvial material may have caused some minor subsidence (because of a negligible excess ice content), there is still nearly 2 m of material missing (i.e. 4 m minus 1.5/2 m). Hence, we assumed there was a 1-2 m pre-existing depression.

We modified the text to make it clearer. For example, we added lake maximum depth (~ 4 m) in the sentence, and we added the following sentence: « Since this silty peat unit is about 1.5-2 m in thickness when still frozen (Fortier and Allard 2004), and since the underlying glaciofluvial unit is ice-poor (thus negligible subsidence upon thaw), there is 1-2 m elevation gap which can be explained by the presence of a preexisting depression. The latter is interpreted as a channel in the glacio-fluvial outwash underlying the silty peat. »

*Update (21 February 2020):*
The new sentence has been modified: « We explain the elevation gap between the maximum lake depth (~ 4 m) and the potential thaw subsidence of the sediments where Gull lake initiated by the presence of a preexisting depression 1-2 m deeper than the surrounding polygonal network. This depression was interpreted as a channel in the glacio-fluvial outwash underlying the silty peat, similar to channels observed today in the glacio-fluvial outwash in glacial valleys of Bylot Island. » (lines 380-384)

*372-373: I'm confused by this part of the conceptual model. Please explain how "the deepest parts of the lake have now almost reached the underlying glacio-fluvial sand." (You also make this statement in lines 427-428.) As ground ice melts and subsidence occurs, the upper sediment layers reduce in thickness, but they typically are not removed. Are you indicating that the upper sediment layers beneath the lake bed are being removed as the lake expands, exposing the glacio-fluvial sands? If so, how is it that you can still see evidence of ice wedge polygon ridges and troughs in the deepest part of the lake bed (lines 363-367)?*

We apologize for the confusion. We were inaccurate in our choice of words. The lake bottom is indeed still covered by silty peat sediments overlying glaciofluvial sands, as seen in the collected cores (Fig. 4). We meant that the 'thawing front' (or the base of the talik) is moving downwards (as thermokarst occurs) and has « now reached the underlying glacio-fluvial sand ». The same reasoning is valid for the statement at the end of section 5.3 (Implications for Arctic carbon dynamics).
We modified these two sentences:
« […] the 'thawing front' (i.e. the base of the talik) has now reached the underlying glacio-fluvial sand […] »
« […] since the the base of the talik has reached the much less organic-rich layer […] ».

**Reviewer #2 (Anonymous)**

*The manuscript "Thermokarst lake development in syngenetic ice-wedge polygon terrain in the Eastern Canadian Arctic (Bylot Island, Nunavut)" presents a careful study of thermokarst lake initiation outside of its main distribution area. Both sedimentation regime and ecology are reconstructed in this highly relevant study from the Canadian high Arctic, a region that is still vastly understudied, mainly due to its remoteness and challenging accessibility. The manuscript is very well written and well structured and presents its findings in a clear and concise way. The study works without an age depth model, but that cannot always be forced, especially in Arctic thermokarst lakes. The way the authors deal with this issue may be the most honest way to present the radiocarbon dates. The dates still give a general indication of the ages of the strata. The authors present a new conceptual model of late Holocene thermokarst lake development. This landscape type and region is indeed strongly underrepresented in the thermokarst literature. I have listed a few general comments, detailed comments and minor edits below and advise to accept this manuscript after minor revisions.*

Thank you for these positive and useful comments. We answered to all the comments (general, detailed and minor edits) made by the reviewer.

General comments

*1. The discussion should focus on/refer to the actual results more obviously. Large parts of the discussion are quite general.*

We modified and added several sentences in the discussion section, as suggested by the reviewer in his/her detailed comments. However, this comment is general and does not focus on specific sections. We would welcome any specific comment, including line numbers, about the discussion. Where exactly could we « refer to [our] actual results more obviously »?

*2. I generally like that there is a section in the discussion dedicated to the wider implications of your findings. This section does, however, need some work still. My main concern here is that you are comparing syngenetic permafrost with Yedoma (which is also syngenetic permafrost). I am not convinced that the difference lies primarily in syngenetic vs. epigenetic permafrost.*

We do not argue that that the difference lies in syngenetic *vs*. epigenetic permafrost. Both our site and Yedoma regions are affected by syngenetic permafrost. See our reply to the next comment just below.

*It is more a question of wetlands vs. non-wetlands. Formerly glaciated terrain often develops into wetlands studded with lakes, but there are also regions which were never glaciated and are rich in lakes and wetlands, e.g. the Arctic coastal plain of Alaskan or its continuation into Canada, or*

*generally Beringian coastal lowlands. To me, the main difference lies in minerogenic vs. organic/peat deposits.*

From a geomorphological perspective, we chose to compare soils that have experienced similar pedogenetic and geomorphic processes, as the processes of soil organic carbon inclusion into permafrost strongly influence their concentration relative to depth (Bockheim 2007; Tarnocai et al. 2009). Since permafrost at Bylot Island developed syngenetically during the Holocene (Fortier and Allard 2004), we chose to compare it to Yedoma deposits, which also developed syngenitically but over the late Pleistocene (e.g., Schirrmeister et al. 2011; Strauss et al. 2017), allowing a comparison of sites with similar permafrost development and pedogenetic/geomorphic history. This is also justified by comparable ground-ice content (in volume) and soil bulk densities (see section 5.3 in the discussion). We are aware that this comparison might not be ideal had we wanted to compare the site with others of similar concentrations, but from a geomorphological and geocryological perspective, they are very similar and highly comparable.

*Also, I am not sure the entire terrace you are studying is homogeneous in its organic matter content. Fortier and Allard, 2004, covered two low-centred ice-wedge polygons from the terrace, in which high organic matter contents can be expected, as these are usually wetlands. Your study looks at one particular thermokarst lake. The findings are relevant, and it is also important to place the finding in a wider context, I am just not too happy with the emphasis on the quantitative comparison.*

Gull lake is located on the highest benches of the ice-wedge polygon terrace of the valley. The ice-wedge polygons studied by Fortier and Allard (2004) are located very close (< 50 m) to Gull Lake and at the same elevation as the lake shore. These polygons showed similar sedimentation rates, ground ice, organic and sediment content. Given the close proximity between Fortier and Allard's study and Gull lake, and the geomorphological similarity between the ice-wedge polygons on this level of the terrace, we are confident that the organic and ground ice contents are similar between these two sites due to similar sedimentary conditions (eolian silt deposition), vegetation type (graminoïds and bryophytes), humidity (wetland) and climate. Several permafrost cores have been drilled in the surroundings, giving comparable values (Godin, unpublished data; Veillette, 2019; see cited references at the end of this report).

*3. It should be made clearer what is new about the conceptual model. This can be done by editing the text only.*

See our reply at the end of this report (last comment about Figure 6).

Detailed comments

*Line 33: "remarkably" sounds a bit weird in this context. Please also reconsider the phrase "circumpolar regions", as thermokarst lakes are strictly speaking most abundant in terrestrial Arctic lowlands.*

We changed 'remarkably' to 'notably'.
We added the phrase 'especially in terrestrial lowlands' in that sentence.

*Lines 42-44: Unglaciated ice-rich terrain is not necessarily Yedoma, it includes icewedge polygon peatlands and lowland thermokarst. Also, if you are categorizing lakes, you might have to be more explicit. Lakes in Yedoma terrain might still be thermokarst lakes, even if they tend to have a different morphology. Not sure this categorization is needed here.*

We agree that detailed lake categorization is not needed here. We just want to specify that 'Yedoma lakes' represent a relatively minor group of lakes, notably deeper and with a different history during the late Pleistocene, compared to the much more abundant lakes developed during the Holocene, in formerly glaciated terrain. We slightly modified the sentence, and added a new one: « However, the vast majority of thermokarst lakes across the Arctic are shallow (a few meters) and were formed in formerly glaciated terrains during the Holocene (Grosse et al., 2013). »

*Lines 50-51: "When thaw depth exceeds the maximum thickness of winter ice cover, [...]" - this is ambiguous. Please rephrase.*

We agree that this was ambiguous. The new sentence is: « When lake depth exceeds the maximum thickness of winter ice cover, water stays unfrozen throughout the year and mean annual lake-bottom temperature remains above 0 °C, resulting in the formation of a talik (thaw bulb) underneath the lake (Burn, 2002) ».

*Line 60: "drawdown" - this might not be the most appropriate term here, especially when you are also using it for lake infilling, could use "lowering" or "decreasing lake depth" instead.*

We are not using 'drawdown' to refer to lake infilling (involving inputs of sediments), but rather refer to increased evaporation, which is really a loss of water volume, thus we had to keep the expression 'drawdown' (i.e. « withdrawal of water from a reservoir »).

*Lines 82-83: your third objective could end with "specifically for syngenetic ice-wedge polygon terrain" or else convince me that your conceptual model is universal.*

We added the phrase « in syngenetic ice-wedge polygon terrain », as suggested.

*Line 99: Did glacier retreat stop for good or is it retreating again now? Also, please give a reference for the date.*

Yes, glaciers are still retreating up in the valley nowadays. This has been documented by Dowdeswell et al. (2007) for the majority of glaciers on Bylot Island. Their study indicates that overall glaciers have retreated from 0.9 to 1.8 km since about 120 years ago, with most retreat occurring between 1958/1961 and 2001.

*Update (21 February 2020):*
We added the following sentence: « Like the majority of glaciers on Bylot Island, the C-79 glacier has recently been retreating up in the valley, at a rate of 0.9 to 1.8 km since about 120 years ago, with most retreat occurring between 1958/1961 and 2001 (Dowdeswell et al., 2007). » (lines 122-124)

*Line 106: not sure "off-shore" is the appropriate term here, it sound like way off the sea shore.*

We assumed that the correct line number referred to is rather 116, and we changed for 'from the central zone' to clarify this point.

*Line 122: Consider indicating that this publication describes the method. It sounds like the results have already been published.*

We added 'as in' in front of the reference, so it's now clearer that it refers to the method.

*Lines 175-176: could refer to the full diatom data set in the database.*

Good suggestion. We added the sentence: « The complete diatom dataset is available in open access (Pienitz et al., 2019). »

*Lines 192-193: Consider indicating that this is data, not a published study. It sounds like the results have already been published.*

We modified the sentence accordingly (« raw data can be found in data repository; Fortier et al., 2019).

*Line 228: Do you trust the age at this depth? It sounds a bit old for a depth of 10cm, and as you dated bulk sediment, how can you be sure this is not, at least in part, relocated old material? Please comment shortly.*

We cannot totally rule out some remobilization of materials, as bulk sediment sample was dated (no organic debris found in that layer). However, we are confident in that age because: 1) short-term dating ($^{210}$Pb) of surface sediments was conducted and showed that at that depth (10cm), there was no more supported $^{210}$Pb, indicating sediments older than 150 years; 2) dating of this layer was also dated in the other core (from 2014), giving the same age.

*Update (21 February 2020):*
We added the following sentence: « According to short-term dating ($^{210}$Pb) of surface sediments above that depth (data not shown), only the first few centimeters appear to be younger than ~ 150 years. » (lines 277-278)

*Line 233: Is "major taxa" a known term? If the 5% abundance in at least one sample (?) is a commonly used standard cutoff, please give a reference. I know this is done to minimize statistical issues arising from extremely low counts and too many zero entries ("not present") in the dataset, but you could consider stating this in the text.*

In this case, the 5% relative abundance in at least one sample cutoff was not based on a statistical criterion (like, for example, for the lower weight assigned to rare species to limit their influence within multivariate statistical approaches). We rather took a practical decision to show within the limited space of the summary diatom diagram (Figure 5) only the dominant and most relevant taxa. We modified the text to make it clearer: « Among these, the 15 most frequently encountered taxa (species or species groups) representing more than 5 % in relative abundance in at least one sample were selected to show major ecological changes that occurred in the past (Fig. 5) ».

*Line 234: "one level" – please specify what you mean here, one zone, one sample or something else?*

It means one sample, and we changed the text accordingly.

*Line 315: You could start this section with "Based on our findings on the geomorphology and palaeolimnology of..." to make it clear that you are talking about the new results rather than about findings from the literature. Or if it is both this study's findings and the results of your lit review, you can say so. This would make it clearer what you have added to the knowledge on thermokarst lake evolution in the region. You can also use active voice in the statement following in line 318 ("we summarized the initial conditions..." or something similar).*

This section 5.2 reports on our new results, while findings from the existing literature are given in the previous section 5.1 (about the Holocene history of the valley). We now start the sentence as suggested (« Based on our findings... »), and we also used the active voice at the beginning of the next paragraph (« We summarized... »).

*Line 323: "must have" – I am not too happy with this absolute phrasing. Please also prove this and give references*

This was also underlined by Reviewer #1. The reasoning is as follows:

In the sediment core from 2015, collected at ~ 4 m depth, we sampled about 0.7 m of silty peat. This unit is currently unfrozen. We know that the surrounding frozen ground of that unit contains over 50 % of ice by volume (Fortier and Allard, 2004). Hence, considering thaw settlement and consolidation, the silty peat layers found in the core must have made at least twice their current thickness when they were still frozen. That makes about 1.5-2 m thick of frozen silty peat before the lake started to form. Even if we assume that the thawing of the underlying glaciofluvial material may have caused some minor subsidence (because of a negligible excess ice content), there is still nearly 2 m of material missing (i.e. 4 m minus 1.5/2 m). Hence, we assumed there was a 1-2 m pre-existing depression.

We modified the text to make it clearer. For example, we added lake maximum depth (~ 4 m) in the sentence, and we added the following sentence: « Since this silty peat unit is about 1.5-2 m in thickness when still frozen (Fortier and Allard 2004), and since the underlying glaciofluvial unit is ice-poor (thus negligible subsidence upon thaw), there is 1-2 m elevation gap which can be explained by the presence of a preexisting depression. The latter is interpreted as a channel in the glacio-fluvial outwash underlying the silty peat. »

*Line 336: Give reference. Also, consider changing "high-centered polygons" to "icewedge polygons" in general, all types of which provide a mosaic of terrestrial and freshwater ecosystems in very close proximity of each other:*
*e.g. Bliss L.C. 1956. A Comparison of Plant Development in Microenvironments of Arctic and Alpine Tundras. Ecological Monographs 26, 303-337, 10.2307/1948544.*
*something newer:*
*from Siberia: De Klerk P., Teltewskoi A., Theuerkauf M. & Joosten H. 2014. Vegetation patterns, pollen deposition and distribution of non-pollen palynomorphs in an ice-wedge polygon near Kytalyk (NE Siberiawith some remarks on Arctic pollen morphology. Polar Biology, 1393-1412, 10.1007/s00300-014-1529-3.*
*From western Canada: Wolter J., Lantuit H., Fritz M., Macias-Fauria M., Myers-Smith I. & Herzschuh U. 2016. Vegetation composition and shrub extent on the Yukon coast, Canada, are strongly linked to ice-wedge polygon degradation. 2016, 10.3402/polar. v35.27489.*
*This is also found in palaeostudies using biological proxies, including diatoms: Fritz M., Wolter J., Rudaya N., Palagushkina O., Nazarova L., Obu J., Rethemeyer J., Lantuit H. & Wetterich S. 2016. Holocene ice-wedge polygon development in northern Yukon permafrost peatlands, Canada. Quaternary Science Reviews, 10.1016/j.quascirev.2016.02.008.*

This sentence is based on field observations. We added such a mention in the text (« […] observable today in the valley »). See the picture below (high-centered polygons).

[Figure]

*Line 352: Is this really typical of thermokarst lakes? See if other references state other accumulation rates, perhaps check Biskaborn, B.K., Herzschuh, U., Bolshiyanov, D. et al. J Paleolimnol (2013) 49: 155. https://doi.org/10.1007/s10933-012-9650-1*
*Klein et al., 2013 - https://doi.org/10.1016/j.palaeo.2013.09.009 or similar.*
*There is slightly more data on carbon accumulation rates in thermokarst lakes (e.g. Anthony, K., Zimov, S., Grosse, G. et al. 'A shift of thermokarst lakes from carbon sources to sinks during the Holocene epoch. Nature 511, 452–456 (2014) doi:10.1038/nature13560*
*Sediment accumulation in thermokarst lakes has been shown to be messy and not at all constant. It can also be fairly high. See for example:*
*Schleusner et al., 2015 doi:10.1111/bor.12084*
*or Lenz et al., 2016 https://doi.org/10.1007/s41063-016-0025-0*
*or Wolter et al., 2017 https://doi.org/10.1177/0959683617708441*

We agree that sedimentation rates in thermokarst lakes can be quite variable and sometimes very high, as the reviewer points out. High accumulation rates are usually associated with active shore erosion and slumping along several meter-high bluffs. This is not the situation at Gull Lake. Although we observed signs of shore erosion such as drifting peat blocks, the shores are less than a meter high and much of the thermokarst activity around the lake occurs as subsidence (see the now-submerged peripheral platform in the accompanying video). Moreover, dissolved organic carbon (DOC) concentrations, as well as nutrients and suspended solids are rather low in this lake, indicating that erosion and slumping is not significant along the shores. Finally, there is no inlet to this lake. For all these reasons, we conclude that accumulation at Gull Lake is rather slow, as measured from sediment traps deployed in thermokarst systems of the sub-Arctic (Coulombe et al. 2016).

*Line 356-358: You are introducing new results here (technically). Perhaps mention this earlier in the manuscript in the appropriate sections (methods/results)? I haven't seen it there. This is not major new data, though, so you might also keep it as it is.*

These data were not planned in the design of our study, and they have been used by colleagues among our extended research group (e.g., Preskienis et al., in review). We included them as a supplement, and as the reviewer points out these are minor data, so we decided to keep the text as it was.

*Also, the results section did not state clearly that an unfrozen zone could not be detected from GPR data.*

In the results section we report that the shallow peripheral platform appears to overlie completely frozen ground: « The signal velocity (> 0.13 m ns$^{-1}$) based on the shape of some hyperbolas suggests that they occur in frozen material. ». There is uncertainty about the deeper central basin, due to signal attenuation. We added a sentence at the end of the paragraph to clarify this point: « Their occurrence at shallow depths beneath the central lake basin suggests that the lake does not possess a deep thawed zone (talik) as is often the case underneath deep water bodies. »

*Lines 372-376: Some questions from my side: Do you mean that the lake cannot become any deeper once it hits the glacio-fluvial sand because of its lower ice content? And do you think it could also stay there without disappearing? Is it certain the lake would drain because of topography or could it also coalesce with the lake next to it? Do you think lake infilling is really an option, as accumulation rates are low and the lake might grow both laterally and vertically in the future?*

In response to these questions:
1) Yes, since the glacio-fluvial unit has a low ground-ice content (limited subsidence upon thaw), the maximum depth of the lake (in its central basin) might not evolve significantly in the future. The stratigraphy under the glacio-fluvial sands is unknown. Marine clay was observed under glacio-fluvial sand a few hundred meters away, on a lower elevation bench of the terrace. It is possible that this layer is present under the glacio-fluvial sands of gull lake. Further deepening (thermokarst) will occur within the peripheral platform.
2) We do not know how long the lake will remain, but we suggest two possible outcomes in the discussion: infilling or drainage. It is worth noting that other lakes were drained (partly or totally) within a 1 km radius of Gull lake and we therefore estimate that the drainage scenario is very likely.
3) Coalescence was observed within the valley among smaller and shallower ponds. However, we also observed direct clues of partial lake drainage in the lake next to Gull Lake (informally named Gull Lake-2 or 'GL-2'; see Fig. 1c). Paleo-shorelines

could be mapped and showed that this particular lake was larger in the past and partly drained. Topographically, it is located between Gull Lake and the proglacial river just north. There is a stream between the two lakes (Gull and GL-2).

4) We do not think that complete « lake infilling is really an option ». As the reviewer points out, sedimentation rates are fairly low, and the lake is growing laterally (peripheral platform). Hence, as we state in the modified sentence: « Some partial infilling might have time to occur, but natural landscape evolution is likely to result in partial lake drainage, as suggested by the presence of erosion gullies in the valley and by evidence on such a partial drainage in a nearby lake ». In short, we think that partial drainage of the lake is the most likely scenario (see next paragraph). Erosion gullies have been observed elsewhere in the valley, and as stated above, paleo-shorelines mapped around the nearby 'Gull Lake -2' show that this is a likely scenario for the future.

*Line 377: Infilling by aquatic and semiaquatic plants is, to my knowledge, more likely in smaller ice-wedge ponds than in lakes. I agree that basins usually fill up with sediments over time, but this might take a very long time. Jorgenson and Shur, 2007, are talking about infilling ponds along the margins of drained lake basins (while large thaw lakes may form in their centers) on the Arctic Coastal Plain of Alaska. The question is the balance between accumulation and decomposition or transport out of the system (e.g. via a stream), or in this case, possibly also lake deepening through additional thaw subsidence. Lake infilling would likely be a very slow process, especially given the low sediment accumulation rates.*

We agree. This is why we start by mentioning lake (partial) infilling as possible, but not as probable as partial drainage (next paragraph). We end our reasoning by this sentence: « Such a partial drainage is likely to happen to Gull Lake in the future, affecting at least the shallow peripheral platform and leaving a residual smaller lake corresponding to the current deeper basin. »

*Line 379: This argument is not super-convincing. Lacustrine sediments normally accumulate in a lake, so that you found them does not necessarily prove terrestrialization.*

We removed that sentence. For a better transition with the following sentence, we added: « We did not observe direct signs of terrestrialization at our study site. »

*Line 391: give reference(s)*

We added a reference (French, 2017).

*405-410: You could state here for which depths the Yedoma TOC contents were calculated, so it is comparable to your findings. Generally, I think you should not extend the carbon contents of the*

*upper 3-5 m to greater depths, as TOC generally decreases strongly below the first meter or so. Also, to compare a point measurement from organic-rich sediment in a relatively small feature in a heterogeneous landscape with averaged values for the entire Yedoma domain is a bit misleading.*

As stated above (reply to general comment #2), our point here is to present a site containing organic-rich syngenetic permafrost of Holocene age (less commonly reported) and to compare it with Yedoma sites that are much more represented in the literature. Results from Holocene syngenetic ice-rich and carbon-rich permafrost is very poorly reported in the literature and it is not, for the moment, possible to compare averaged Yedoma with Holocene values such as at our site.

*Line 415: Please avoid citing articles in review, as they are not available for checking.*

We expect that the Tank et al. (in review) paper will be released by the time that our manuscript is (hopefully) published.
Note to the Editors: if this is not the case, then we can remove this citation and replace it with another one.

*Update (21 February 2020):*
We removed the Tank et al. citation; it is still under review.

*Lines 424-427: This could be 2 sentences. Be extra careful in your phrasing here: "the entire Yedoma complex" sounds like you mean all of the Yedoma there is (its entire area).*

We modified the sentence accordingly ('entire' removed): « In other words, to obtain the same amount of organic carbon released from thawing of the upper 3 m permafrost terrace on Bylot Island, an equivalent of 30 m of Yedoma complex would have to thaw, which is extremely unlikely in the foreseeable future ».

*Also, as commented above, be careful with that comparison. It is valid to say that there are landscapes in the arctic that contain more organic-rich sediments than found on average in Yedoma, but this quantifying comparison is going too far for my taste, see my general comment above. Consider citing GHG emissions from Arctic wetlands. And how do your findings relate to findings from other Arctic wetlands or other lowcentered polygon fields? You could for example compare your findings to those from ice-wedge polygon wetlands in the western Canadian Arctic, i.e. on the Yukon Coastal Plain, the eastern part of which used to be glaciated, or on the Tuktoyaktuk Peninsula.*

We want to draw attention to syngenetic permafrost in organic-rich Holocene deposits, which is under-represented in the literature. After all, Yedoma complexes are not that carbon-rich, and large areas where peat accumulation and burial are important show

much greater organic carbon contents (such as in Bylot). See our reply to the general comment above.

If the Editors agree with that suggestion, we could add to this paragraph a comparison to some other ice-wedge polygons sites (e.g., from the Yukon Coastal Plains) as suggested by the reviewer.

*Update (21 February 2020):*
We added a sentence and modified the next one (lines 534-537; 555). See our reply to the Editor's comment above (p. 2).

*Line 439: Thermokarst lake cycles would take far longer to develop. There simply wasn't enough time for that. The study design was thus not suitable for testing whether thermokarst lakes develop cyclically or unidirectionally.*

Agreed, and we removed this sentence.

*Lines 437-438: Not all Pleistocene-age permafrost deposits are Yedoma.*

But Yedoma deposits are all of Pleistocene age, so we 'flipped' the sentence around to make it clear: « […] which is dominated by Yedoma deposits (Pleistocene-age ice-rich permafrost) ».

*Lines 441-444: "regardless of climate" – that might be a bit too much. In your next sentence you rightly state that precipitation (which is also climate) and snow distribution (which has a geomorphological component) are more important than temperature development. Stick with that.*

We changed the sentence accordingly (« […] regardless of air temperatures. »).

Technical edits

*line 43: replace "since" with "in"*

Done.

*line 54 and elsewhere: is the "in prep." manuscript published now? If not, omit reference.*

It has been submitted and is now 'in review'. We modified the text here and elsewhere.

*Line 81: omit "a" before "syngenetic ice-wedge polygon terrain"*

Done.

*Line 87: developed into*

Done.

*Line 91: total precipitations -> a total precipitation*

Done.

*Line 93: perhaps better to separate this insert with commas instead of brackets*

Done.

*line 105: "before 3700 years ago" - sometime before? Just before?*

This is a minimum age (14C). We changed to 'at least' 3700 years ago.

*Line 128: not sure one can "conduct" survey lines?*

Agreed, we changed to 'done'.

*Line 198: the lake bottom*

Done.

*Line 218: "..., which are both dominated by peat"?*

Done.

*Line 254: "in average" should be "on average"*

Done.

*Line 268: separate the insert however by commas (was, however, strongly)*

Done.

*Line 269: Better use "entire" instead of "whole"?*

Done.

*Line 297: Perhaps better to say "During" or "At the beginning of the" late Holocene*

Done ('During the late Holocene…').

*Line 323: "found at the lake bottom"*

Done.

Line 337-338: "were a significant source of latent heat to extract in autumn" – this needs some rephrasing. I do not much like the use of the word significant outside when not talking about statistical significance. And I do not quite understand the word "extract" in this context.

We removed 'significant' and the allusion to heat 'extraction'. The modified sentence now reads as follows: « In autumn, heat loss from these small water bodies to the atmosphere and subsequent phase change of water to ice delayed the freezing front propagation in the underlying ground (Kokelj and Jorgenson, 2013) (stage 2; Fig. 6c) ».

*Line 354: the lake bottom*

Done.

*Line 386: water balances -> water balance*

Done.

*Line 392: tapping*

Done.

*Line 408: "presents slightly over" could be "contains more than"*

Done.

*Line 411: "are comparable to other circumpolar regions"?*

Done.

*Line 418: formerly glaciated terrain ?*

We are talking about numerous sites here, so we need to keep the plural form (terrains).

*Line 442: "self-enhancing"?*

Done.

*Figure 5: Is it possible to add ecological interpretation/groups on top of the taxa? That might help readers.*

This is unfortunately not possible because species or species groups are presented according to their occurrence at the site along with ecological/habitat changes through time (from bottom to top of the sediment core).

*Figure 6: Is this conceptual model really new? It looks a lot like the existing models. The only immediate difference I see are the ponds forming on top of the ice wedges instead of between ice-wedge ridges. This might be because of climatic warming and subsequent ice-wedge degradation. How can you prove that the ponds were on top of the ice wedges? Is there a difference in diatom flora between intrapolygonal and interpolygonal ponds?*

One major difference between our model and previously published models is that ground ice melting and the initiation of thermokarst start from the top of ice wedges, either at ice wedge junctions or along longitudinal segments. In previously published models thermokarst initiated in the center of ice wedge polygons, which is counter-intuitive since the ground ice content is much lower than below ice wedge ridges (close to 100% ice). We attached a field photo (below) showing ice wedge degradation with intact polygon center. This situation was common at our study site. Deep ponds at ice-wedges intersections are the weak spots in ice-wedge systems that become especially vulnerable with warming climate. Numerous observations confirm that ice-wedge thermokarst commonly starts at these locations and continues rapidly along ice-wedge troughs. These

degraded ice wedges are usually covered by longitudinal ponds of various depth. These 'collapsed trough ponds' eventually merge together and sometimes merge with small ponds in the middle of polygons.

[Figure]

Another novel element of our model is the fact that lake initiation started in late Holocene sediment during a colder climate ('Neo-glacial'), mostly driven by natural landscape evolution and the strong impact of snow accumulation in a pre-existing topographical depression. We added the following sentence in the conclusions to make it more obvious: « Moreover, this model explains the early formation (inception) of a thermokarst lake during a cooling climatic trend (the 'Neo-glacial), underscoring the importance of natural landscape dynamics over temperature only. »

About the last question (diatom species, intra- *vs*. interpolygonal ponds):
No, it is not possible to tease out intrapolygonal *vs*. interpolygonal pond environments with diatoms. The difference is too subtle or non-existent in terms of substrate or habitat.

Cited references

Abnizova, A., Siemens, J., Langer, M., and Boike, J.: Small ponds with major impact: The relevance of ponds and lakes in permafrost landscapes to carbon dioxide emissions, Global Biogeochemical Cycles, 26, GB2041, doi: 10.1029/2011gb004237, 2012.

Bockheim, J.: Importance of Cryoturbation in Redistributing Organic Carbon in Permafrost-Affected Soils, Soil Science Society of America Journal, 71(4), 1335-1342, 10.2136/sssaj2006.0414N, 2007.

Dowdeswell, E. K., Dowdeswell, J. A., Cawkwell, F.: On the glaciers of Bylot Island, Nunavut, Arctic Canada, Arctic, Antarctic and Alpine Research, 39 (3), 402-411, 10.1657/1523-0430(05-123)[DOWDESWELL]2.0.CO;2, 2007.

Fortier, D., and Allard, M.: Late Holocene syngenetic ice-wedge polygons development, Bylot Island, Canadian Arctic Archipelago, Canadian Journal of Earth Sciences, 41, 997-1012, 10.1139/e04-031, 2004.

French, H. M.: The periglacial environment, 4th ed., John Wiley & Sons, Chichester (UK), 515 pp., 2017.

Schirrmeister, L., Kunitsky, V., Grosse, G., Wetterich, S., Meyer, H., Schwamborn, G., Babiy, O., Derevyagin, A., and Siegert, C.: Sedimentary characteristics and origin of the Late Pleistocene Ice Complex on north-east Siberian Arctic coastal lowlands and islands – A review, Quaternary International, 241, 3-25, 10.1016/j.quaint.2010.04.004, 2011.

Strauss, J., Schirrmeister, L., Grosse, G., Fortier, D., Hugelius, G., Knoblauch, C., Romanovsky, V., Schädel, C., Schneider von Deimling, T., Schuur, E. A. G., Shmelev, D., Ulrich, M., and Veremeeva, A.: Deep Yedoma permafrost: A synthesis of depositional characteristics and carbon vulnerability, Earth-Science Reviews, 172, 75-86, 10.1016/j.earscirev.2017.07.007, 2017.

Tarnocai, C., Canadell, J. G., Schuur, E. A. G., Kuhry, P., Mazhitova, G., and Zimov, S.: Soil organic carbon pools in the northern circumpolar permafrost region, Global Biogeochemical Cycles, 23, GB2023, 10.1029/2008GB003327, 2009.

Veillette, A.: Stabilisation du paysage périglaciaire suite à un épisode de ravinement par thermo-érosion : implication pour la structure et la stabilité thermique du pergélisol de surface, MSc thesis, Dept. of Geography, Université de Montréal, 2019.

[revised manuscript text omitted]

---

## Editor Decision (ED1)

Dear F. Bouchard et al.,

Thank you for your careful consideration of the reviewer's concerns. It is clear from the comments that the reviewers thought quite favourably of your manuscript, and provided many helpful comments that I think will improve the final paper. I would like you to please submit a revised manuscript for me to review.

Reviewer 1 made many helpful and detailed comments. Most appear to be easily dealt with, and I have just a few follow up questions that I hope you can clarify at this stage.

> Line 99: Please ensure there is a commensurate change in the text to reflect your response to the comment.

> Line 228: Again, please ensure that there is a commensurate change in the text to reflect your response to the comment.

> Lines 356-358: If these are results. It may be that these data were not in the original study design, and yet they were not produced by accident. I would like you to please include the results in the results section and discuss here. Or, if the data are already published in Preskienis et al., then you may want to keep here as a part of the discussion of your results in the context of other work. There needs to be a clear line between your contribution and that of others in the extended research group. Somebody needs to take responsibility for the data and explain its origins.

> Line: 415: Here and elsewhere, please see if you can replace such articles "in review" with materials that have been published. Everybody wants to be able to follow up on referenced statements, and this is not possible for materials in review (that may not get published). Please replace those "in preparation" with an alternative.

> Lines 424-427: Please do include a few sentences that compare your site to other ice-wedge polygon sites.

> Figure 6: If the reviewer was not convinced as to the novelty of your conceptual model, you will need to think carefully about it. What is new, and what is in agreement with existing models. I think that you have an opportunity to point out that models that suggest that thermokarst initiates from the low polygon is not in agreement with what we see in the field. There is a lot of literature on thermokarst initiation at the ice wedges (e.g., Mackay 2000, Figure 3), and your model agrees with it.

Reviewer 2 also provided several helpful comments. The only outstanding issue is with respect to the first general comment. R2 would like you to be more pointed in your discussion. Keeping in mind that you have addressed the detailed comments, please review your discussion carefully and see if you can tease out a few more subtleties that draw specifically from your work. It is these non-general details that will help increase the impact of this work, and distinguish it from other research.

Best regards,

Peter

Reference

Mackay JR. 2000. Thermally induced movements in ice-wedge polygons, western Arctic coast: A long-term study. *Géographie physique Quaternaire* **54**: 41-68.

---

## Author Response (AR2)

We thank the Editor for this latest round of review. Below are our responses to the comments and questions. The original comments are shown in *grey (italics, smaller font)*, and our responses are presented in black (normal font). Please note that **the line numbers mentioned in our responses refer to the marked-up version** ('track changes' mode), submitted alongside this letter (thus not the 'clean' revised manuscript).
* * *
**Editor (Peter Morse)**
Received: 06 March 2020

*Thank you for your revised manuscript. You treated the reviewer's comments well. I have read through the paper again and there are some important details that need your attention to sort out. I don't think that the revisions will substantively change your discussion and conclusions, but I hope that the suggested changes will make certain aspects of the manuscript more clear to the reader, and will help showcase some of the novelty. Detailed comments are in the attached pdf.*

Thank you. We have prepared a revised manuscript and we submitted a marked-up version accompanying this letter. We made all the minor/editorial/grammatical changes suggested, and below is a point-by-point reply to the more substantial comments, questions, and suggestions.

*P2, L51. Isn't snow a part of the precipitation regime and thus climate, driver 1?*

We changed « climate » to « increased air and ground temperatures », so that it is now clear that snow is a separate driver from temperature alone (L53-54).

*P2, L52-54. Snow is a freezing season phenomenon, promoting thaw by preventing ground cooling. It does not cause warming, per se.*

Agreed. We changed the text as suggested (L56-57).

*P2, L54. This link between snow and increased thaw depth the following summer has been demonstrated in the western Canadian Arctic and Russia. e.g., Frauenfeld et al. 2004, JGR, 109: D05101; Morse et al. 2012, CJES, doi:10.1139/E2012-012.*

We added the references (L57).

*P3, L83. here do you mean "climate temperature" or temperature and precipitation?*

We changed « climate » to « increased temperatures » (L93).

*P3, L96. At, @160 km away, it is not really nearby. Delete.*

We removed « nearby » and inserted the suggested phrase. It now reads « Based on the 1980-2010 climate normals from the closest meteorological station located in the village of Mittimatalik (Pond Inlet) on Baffin Island, […] » (L105-106).

*P4, L106-107. What is the rate? this is the distance. km y-1, or decade? Instead, please just state the year.*

Agreed, this is not a rate, but rather a distance. We removed this word and modified the sentence as follows: « […] the C-79 glacier has recently been retreating from 0.9 to 1.8 km up the valley since the early 20th century […] ». (L133-134)

*P5, L149-151. Why did you stop each core? Did you advance the corer to the point of refusal? If so, was refusal due to resistant stratigraphic unit (sand/gravel), or frozen ground, or impossible to tell?*

In both cases, the coring was too difficult once the plastic tube was penetrating the sand/gravel unit (underlying the peaty layers, i.e. Unit 1 in Fig. 4). We were able to sample only a few cm of this material at the base of the cores. It is not possible to tell if the resistance was due to a coarse and compact stratigraphic unit (sand/gravel), or because of frozen material. Nevertheless, we added the following sentence: « Coring was stopped when the tube could no more penetrate into the sediments (coarse sands and gravels). » (L187-188).

*P5, L153. Minimize? Can this be totally prevented?*

It probably cannot. We changed « prevented » to « minimize » (L191).

*P5, L155. After this dewatering, was this supernatant also removed, otherwise there can be mixing again, no?*

We added the phrase « after which the supernatant was removed ». (L192-193).

*P7, L212. Probably too shallow, but are any of these potentially the bases of ice wedges? Bode et al., 2008, identified these in GPR data.*

Based on the polarity of the reflections, it appears that these reflectors are linked to the transition from an unfrozen to a frozen medium, hence the identification as top of ice wedges. Furthermore, the spacing between 'ice wedge reflections' is similar to ice wedge spacing out of the lake on the terrace. The coarse (low) resolution and the weakness of the signal would not permit the identification of the bottom of the ice wedges. A stronger signal penetration would be needed, as well as a uniform stratigraphy and a relatively flat surface topography.

*P8, L245-247. New text. $^{210}$Pb is not mentioned in methods, and not data presented. Include a line or two in methods on $^{210}$Pb and either present date in this manuscript or cite the source. Or you will need to delete this.*

We deleted this sentence (L300).

*P11, L329-330. You can't really say that this is a transient layer. Permafrost is developing in a syngentic setting. Permafrost aggradation in "new" surficial material is not transient but simply aggradational. Are the ice wedges truncated or did they just stop cracking for a while? If there are no new ice-veins then the latter cannot be totally ruled out. In the context of the paper, no need to include this phrase. I would delete.*

We agree that in syngenetic permafrost, ice aggrades as the surface rises due to sedimentation. Generally speaking, the transient layer forms at the junction between the active layer and permafrost of epigenetic or syngenetic origin (Shur et al. 2005). The transient layer is associated with the fluctuation of the thawing depth on decadal timescale. This fluctuation creates ice aggradation. Usually, the variation of thaw depth is larger than the accumulation rates in syngenetic permafrost, and this explains why a transient layer is very often present over syngenetic ice wedges. We consider that this point is novel and useful for permafrost/thermokarst modeling and we suggest keeping it. If the Editor wishes that we include a bit more explanations in the text (1-2 sentences) about this specific issue, we can do that.

*P11, L338. The wedges probably do not extend through the clays.*

Exactly. We believe that they reach only the upper clay layers, based on previous studies around Gull Lake, but probably do not extend through the whole unit (Fortier and Allard, 2004). The thickness of clays observed elsewhere nearby on the terrace is unknown.

*P11, L342. Please verify the thickness of the thaw-susceptible sediments. I just entered a range as a place holder.*

The thickness is mentioned earlier in the paragraph (~ 1.5-2.0 when frozen; ~ 0.7 m when thawed) (L402-403).

*P11, L344. Why did the channel not fill in with eolian materials and/or peat? Can you rule out any eolian component of the landscape evolution? Can the depression be related to eolian erosion? It is harder to have very high confidence in the presented model as the interpreted stratigraphy is not shown well in the of the GPR interpretations, just the schematic in the conceptual diagram. Fig. 3 just shows a few arrows in the upper part, but the stratigraphy is not shown in the following 3 interpreted lines.*

The glacio-fluvial outwash channel dried out following isostatic uplift of the valley floor following deglaciation (Fortier and Allard, 2004). Eolian silts and fine sands sedimented in the channel before and during peat accumulation as can be observed in the CT-Scans. We therefore rule out that the depression was formed by eolian erosion. We have high confidence in the interpreted stratigraphy for the portion covered by the cores. Below, we had to use GPR to image the stratigraphy and therefore we have a lower level of confidence. However, our careful interpretations are based and backed by previous cryostratigraphic studies in the vicinity (a few 100 m to < 1 km away from Gull Lake). We added the interpreted contact between the silt/peat unit and the glacio-fluvial sands and gravels on Figure 3 (interpreted lines).

*P12, L364. There is a gap in logic between the previous sentence and this one. Delayed propagation leads to what? Talik development? prevents ground cooling, promoting deeper thaw. More thaw leads to coalescence? Please make the links for the reader.*

For a better transition, we added the following sentence: « Hence, this resulted in more efficient and deeper thaw during the following summer. » (L444-445)

*P13, L387. This is like the basin morphology of tundra lakes at Richards Island (Burn, 2002).*

We added the Burn (2002) reference (L491).

*P13, L391-393. Are any of these verified as ice wedges?*

Not directly beneath the lake basin, but around the lake, in frozen material, yes (Fortier and Allard, 2004). We added this reference (L497).

*P13, L394. Bode et al., 2008 resolved ice wedge bases, are any of the reflectors the base of ice wedges? How do you rule this out? some of the reflectors look like they are below ice wedge tops.*

*Is there something that you can add to your methods to make it a bit more clear what the GPR will/will not resolve?*
*Bode et al., 2008, Estimation of ice wedge volume in the Big Lake area, Mackenzie Delta, NWT, Canada, in Proceedings of the 9th International Conference on Permafrost, pp. 131–136.*

The deep row of reflectors in the first 80 m of Line 12 represent the upper limit of a coarse stratigraphic unit (hence the series of reflectors). It appears as lined-up strong point reflectors, which are often overwhelming the still visible linear reflection occurring on the stratigraphic contact. The depth of this layer beneath the soil surface, assuming a signal travel velocity of 0.13 m ns$^{-1}$, is around 4 m, which is consistent with the stratigraphy presented in other studies (Allard, 1996; Fortier and Allard, 2004). According to the same studies, the bottom of ice wedges is located at about 8 m. Please refer also to the interpretation section, lines 493-501 for discussion points. We have no indication that these should correspond to the bottom of ice wedges. We have added a few words on vertical resolution of the GPR signal in the methods section (L175-176), but the GPR signal will typically permit the identification of units or objects whose presence is already known or suspected.

*P13, L398-402. The scenarios shown in the figure are not realistic. The figure should match these scenarios of disappearance. See the figure for specific comments.*

See our answers to the specific comments about the figure.

*P13, L398-399. Where is the thaw front now? Can you detect this from the GPR? How do you know conclusively that it has not already progressed into the sands and gravels? Your figure 6d shows about 1-2 m of Gyttja /Thawed silt/peat, over sand and gravel. Presumably, your cores advanced to the point of refusal. But was this refusal due to frozen ground or reaching the glaciofluvial sand/gravel? You should try to resolve this question. Perhaps with thermal modeling?*

It is possible to tell that the upper portion of the sand and gravel unit underlying the peaty silt unit was thawed at the time of coring as we did observe thawed sands and gravels at the base of the cores (Fig. 4). It is not possible to determine, however, if sediment recovery was not possible further down due to the type of sediment (thawed and sand and gravel are very difficult to core and recover) or refusal due to the presence of frozen ground. As far as the GPR signal is concerned, in our case, it is not possible to determine if a reflection is created by the frozen/unfrozen boundary without lake sediment probing so we prefer to be careful with this type of interpretation. Thermal modeling could help obtaining some insights about the unfrozen/frozen boundary, but we consider that this is well beyond the scope of our paper.

*P13, L412-413. This can occur to any lake, not just thermokarst lakes.*

We removed « thermokarst » in that sentence (L530).

*P14, L426-427. TC readers want to see the data/findings/evidence presented. You show a GeoEye-1 image of Gull Lake, and GL-2 is right beside it. You can just show GL-2 in a figure and the reader will believe you. Then you don't have to say "unpublished data".*

We produced a new figure (Fig. 7), which shows former lake shores, pingos and less defined polygonal networks in former lakebeds. We also added this information in the figure caption (p. 34).

*P14, L445. For a general reference, you could cite Williams and Smith, 1989.*

We added the reference (L580).

*P14, L446. Biskaborn et al. do not consider thaw of ice-rich permafrost so deep thaw is not realistic at your site in their model time frame. Also, your paper indicates that in this setting, thaw is not that deep, even if there is ponded water.*

We removed the allusion to deep (several meters) thawing. The sentence now reads « Nevertheless, if future climate change results in even more widespread thaw of ice-rich permafrost (Biskaborn et al., 2019), […] » (L581).

*P15, L456-457. Please show this in the results, otherwise this finding is not well substantiated.*

This is not a direct observation (i.e. we could not clearly identify the thaw front on the GPR profiles). We conclude this based on 1) the fact that the collected sediment cores reached the glacio-fluvial sands (observed at the base of the cores), and 2) the similar $^{14}$C age – between 3.5. and 3.8 kyr BP – reported from the base of the silty peat unit (Fortier and Allard, 2004) and the methane bubbles sampled near the coring site (Bouchard et al., 2015). We modified the sentence accordingly: « For the specific case of Gull Lake, as the base of the talik has now reached the organic-poor layer of glacio-fluvial sands, as shown by $^{14}$C maximum age of methane corresponding to the maximum age of the permafrost terrace (Bouchard et al., 2015; Fortier and Allard, 2004), future emissions from the deep basin are likely to slow down, although lateral expansion will likely fuel emissions from the current peripheral platform. » (L589-592).

*P15, L465-466. Not clear. What is underrepresented, what is dominated by Yedoma. Please revise.*

Models of lake inception and development for formerly glaciated terrains in syngenetic permafrost landscapes are underrepresented in the literature, while similar paleoenvironmental reconstructions are much more abundant for Yedoma regions. We modified the text, which now reads: « Paleoenvironmental reconstructions of formerly glaciated syngenetic permafrost landscapes are currently underrepresented in the thermokarst lake literature, which is dominated by Yedoma deposits (Pleistocene-age ice-rich permafrost). » (L600-602).

*P15, L470. Please take the opportunity to highlight the influence of landscape history on the creation of this thermokarst lake. What about the paleo-topography? That is, the pre-exising depression that may be due to an old channel? This is a major component of the conceptual model, but it is not mentioned here.*

We added the following sentence: « The existence of a pre-existing depression (abandoned glacio-fluvial outwash channel), collecting snow and meltwater and therefore affecting ground surface temperatures, underscores the control of paleo-topography on permafrost landscape evolution. » (L609-611).

*P26, Table 1. Tables simply have a 1 sentence title. The remainder should be included as a note.*

We moved the second sentence below the table, as a note (p. 27).

*Also, you should mention in your methods that you enriched your 14C dataset with additional dates determined in previous studies.*

We added a sentence at the end of the paragraph presenting chronological analyses (3.3): « Supplementary $^{14}$C dates available for the surrounding frozen peat deposits, some of them not yet published, were also compiled and added to the dataset (Allard, 1996; Ellis and Rochefort, 2004; 2006; Fortier et al., 2006; 2020). » (L219-221).

*P26, Table 1 (Veillette reference). Not a co-author, not acknowledged, not even a full name. The international reader cannot track the responsible person down. The data are published here, so best to move to Bouchard et al., and discuss with Veillette how to acknowledge the contribution.*

We included the date in the table as 'Bouchard et al.'. We submitted a new dataset compiling unpublished $^{14}$C dates from Bylot Island (Fortier et al., 2020). This new dataset is not mentioned in the 'Data availability' section (L688-689) and in the reference list (L851-852).

*P26, Table 1 (Fortier reference). Fortier is a co-author, so include in Bouchard et al., and work the acquisition into the methods section.*

This is what we did; we added these dates into the table as 'Bouchard et al.', and we prepared a related dataset. See our response to the previous comment.

*P28, Figure 2 (dashed line). Much too faint. Also indicate what the blue line is, and make the blue line thicker, e.g, "The lake limit is delineated by a blue polygon."*

We modified the figure as suggested, and we added a mention to the blue line (p. 29).

*P29, Figure 3. Please describe what the lower figures show. There is currently no description. E. G., what are the triangles? Presumably high returns, but you need to say so.*

*Why are some of the triangles stacked, are they top and bottom of a wedge, or just coincidence?*

Stacked triangles are interpreted as the strong returns at the top of the glacio-fluvial layer composed of sands and gravels (see interpretation section, lines 493-501). They could have been presented as a line, but since they partly obscure the linear reflector, we simply identified them as reflectors. As explained in a previous comment, they are too shallow to be interpreted as the bottom of ice wedges.

*It would be great so see an example of the interpreted stratigraphy overlain on the GPR data.*

This would have been ideal, but there are no stratigraphic details identifiable with the 50 Mhz GPR antennas, except for the glacio-fluvial unit on the edge of the frozen lake. The vertical resolution of the GPR frequency is simply too low (around 0.8 to 1 m) to provide such details, and a greater resolution (100 – 200 Mhz) would not have penetrated enough and would have been completely attenuated by the water.

*Please show the GPR data and the schematics at the same scale. Position distance scale should be the same (450 m) for all lines and the GPR, and the time/depth scales should cover the same domain. Lines 13 and 14 are shorter than line 12 and should appear so.*

We have modified the figure accordingly (p. 30).

*Where is the sand/gravel layer beneath the central basin? the conceptual model draws distinct stratigraphic bounds that are not readily apparent here.*

This layer is not clearly visible below the basin on the GPR survey, as much of the signal is scattered by strong reflectors (ice-water boundary and water-sediment boundary) and attenuated by travel through water. Because this layer was sampled by coring, we inferred that it is present beneath the central basin.

*P29, Figure 3 (caption). For what line? 12?*

Yes, line 12. We added this information (L1060, p. 30).

*P29, Figure 3. Please make the arrows more bold so that they are easier to see. Perhaps add a black outline.*

Modifications have been made to the figure.

*P30, Figure 4. Many of the fonts used for labels and mark up are too small, and I'm viewing at 125%.*

We enlarged the fonts in this figure (p. 31).

*P31, Figure 5. Please include a sentence indicating that the relative Abundance scale varies by taxa. This figure also has a lot of too small fonts.*

We included this sentence, as requested. We also enlarged the fonts in the figure (p. 32).

*P32, Figure 6 (b – Stage 1). The thickness of the glacio-fluvial deposit is different than above.*

We modified the figure accordingly (p. 33).

*P32, Figure 6 (d – Stage 3). Further subsidence is not possible because thaw has reached sands and gravels that are not ice rich. Does this error affect the description of the conceptual model in the text?*

Further subsidence should rather be involved, in fact, between stages 2 and 3 (L443-452). We modified the figure accordingly, using « **subsidence +** » in ice-rich material (silty peat layers; Fig. 6c) and « **subsidence -** » in ice-poor material (sands/gravels; Fig. 6d). Thawing of the ice-poor sands and gravels would result in thaw consolidation and some subsidence. Because an ice-rich clay layer is likely present beneath the sands and gravels, subsidence should increase when this layer will thaw in the future, assuming no lake drainage.

*P32, Figure 6 (e – Stage 4a). Stage 4a shows subsidence and ponding, but the main text in the manuscript is about terrestrialization.*

Stage 4a rather shows lake infilling (thicker gyttja), i.e. partial terrestrialization. We slightly modified the figure to make it clearer.

*P32, Figure 6 (e – Stage 4a). Where did the sand and gravel go? This apparent loss of sand and gravel is not the scenario discussed in the text.*

This is our mistake, there should still be some sands and gravels (as observed in the sediment cores). We modified the figure.

*P32, Figure 6 (f – Stage 4b). Truncated ice wedges can reactivate, but why were these wedges truncated here if they did not thaw in other Stages? Stage 3 shows them near the active layer. Stage 4a shows them under water well into the future and still not truncated this much.*

Ice wedges become truncated as of stage 1 when pools start to form on the ridges. This truncation progresses in the following stages, as the thaw front progresses. At stage 3, the active layer is in contact with the ice wedges, which indicates advanced thermokarst. The scale of the figures does not allow to appropriately represent the truncation. Nevertheless, we added a mention to ice-wedge truncation in the caption (L1093, L1098; p. 33).

*P32, Figure 6 (f – Stage 4b). Again, this subsidence is not possible.*

See above. We modified the figure accordingly. We also added 'frost heave' along the exposed shores (paleo-shorelines).

[revised manuscript text omitted]

---

## Author Response (AR3)

We thank the Editor for this latest round of review. Below are our responses to the comments and questions. The original comments are shown in *grey (italics, smaller font)*, and our responses are presented in black (normal font). Please note that **the line numbers mentioned in our responses refer to the marked-up version** ('track changes' mode), submitted alongside this letter (thus not the 'clean' revised manuscript).
* * *
**Editor (Peter Morse)**
Received: 20 May 2020 (annotated PDF)

*Almost there! Many many improvements! Thank you. There are still some things that need to be addressed before publication. The majority are minor, but Figure 6 still needs some careful re-drafting. It really all comes down to making more use of the glacio-fluvial outwash channel in the figure so that the cartoon matches up better with your conceptual model. I would like to have a look at the figure before sending to publication.*

Thank you. Again, we made all the minor/editorial/grammatical changes suggested, and below is a point-by-point reply to the more substantial comments, questions, and suggestions. We re-drafted Figure 6 as requested. Please see details below.

*P3, L97. (inserted) « ~ 160 km away »*

We double-checked, and this value is actually a 'round-trip' distance. The real distance between Mittimatalik (Pond Inlet) and the study area on Bylot is around 80 km (L98).

*P11, L338. Allard (1996, p. 210) says that in section the ice wedges reach the active layer, and are not truncated somewhere below it.*

We slightly modified the text: «… with the top of ice-wedges coinciding with the base of the active layer. » (L353).

*P11, L338. Delete. It reads like a word or two are missing, and the phrase is unnecessary. The transient zone is not discussed anywhere else in the manuscript and is not a part of your conceptual model, so it doesn't really need to be invoked.*
*[…]*
*Sorry for going on about this, but I hope you see my points. If you do need to invoke the transient layer here, you need to support it with some observations and explanation.*

We deleted the mention to the transient layer (L353).

*P11, L338-339. Shur et al., 2005 do not provide metrics on these ice wedges.*

We removed this reference (L353).

*P11, L349. Is there any subsidence potential? I would replace "low" with "little". The sands and gravels have a high density as you indicate because they are well packed fluvial sediments, and fortier and Allard (2004) say they are ice-poor, so it is not clear how there could be much if any thaw consolidation. Sands and gravels are generally considered thaw stable.*

We replaced 'low' with 'little' (L364).

*P11, L351. When the reader looks at Figure 6a, this depression of the surface by up to 2 m is not obvious.*

We re-drafted Figure 6, including a more obvious depression in Fig. 6a (see below).

*P11, L352. Do you mean the depression at the surface is interpreted as an expression of a channel at depth? The channel is down in the sands/gravel, and is infilled by silt/peat.*

*The it is the remnant depression that you invoke as initiating thaw subsidence.*

*If Figure 6 showed a much more obvious depression in the sands/gravels, and this was filled with silt/peat, then there would be a higher overall potential for thaw subsidence, even if the sediment accumulation didn't completely level things out but left a remnant depression. More silt/peat means more thaw consolidation.*

Thank you for this modification. We re-drafted Figure 6 to take this comment into account. The depression in sands/gravels (glacio-fluvial unit) is now more obvious.

*P13, L390-392. The bumpy bathymetry, with variable contact with winter ice cover should be reflected by the active layer. Can you show this somehow in the model? You show a bumpy bathymetry.*

We re-drafted Figure 6, including a slightly 'wavy' active layer in Fig. 6d.

*P16, L493. Please update dataset that you recently mentioned in an email.*

We added the new dataset (Fortier et al., 2020) (L539-540).

*P16, L510. Please verify this link. I tried to look at the video and my browser timed out trying to access it.*

We double-checked with two different browsers (Safari and Firefox), and the link is indeed working (last access 15 June 2020).

*P27, L820. Please use a proper citation and add the data to the References and to the Data availability section.*
*Further, when you follow this link, Nordicana D indicates that this DOI does not exist or is unavailable.*

We modified the text (L862) and added this new dataset in the appropriate sections (References and Data availability). The Nordicana-D DOI should now work.

*P29, Figure 2. Presumably the colours shown on the lines also represent depth. How come the colours shown on the survey lines are not the same as those shown in the Bathymetric inset? For example, the inset shows a lot of dark blue where line 13 crosses, but line 13 is mostly green. Same for line 12.*

The scale is the same, however the values are slightly different. The bathymetry (inset) was created by interpolating sonar data, while the depths on the GPR lines represent GPR depths. When compared, the RMSE = 0.39 m.

*P30, L841. Are these data under review or published? Please do not include hyperlinks in captions. Remove and add in the proper citation. Please ensure the reference is included in the reference list and referred to in the "Data availability" section.*

These data (Fortier et al., 2019) have been reviewed and are now accessible online. We removed the hyperlink in the caption and we now refer to these data in the appropriate sections (References and Data availability).

*P31, L846-847. Please do not include hyperlinks in captions. Remove and add in the proper citation. Please ensure the reference is included in the reference list and referred to in the "Data availability" section.*

These data (Fortier and Bouchard, 2019a; 2019b) have been reviewed and are now accessible online. We removed the hyperlinks in the caption and we now refer to these data in the appropriate sections (References and Data availability).

*P32, L853. Please do not include hyperlinks in captions. Remove and add in the proper citation. Please ensure the reference is included in the reference list and referred to in the "Data availability" section.*

These data (Pienitz et al., 2019) have been reviewed and are now accessible online. We removed the hyperlink in the caption and we now refer to these data in the appropriate sections (References and Data availability).

*P33, Figure 6. This set of figures still needs work. A problem with this set of figures is that the base figure, "a", is not propagated through the rest of the model. The wedges sometimes change from frame to frame (notwithstanding truncation), and the glacio-fluvial sediments change from Stage 1 to 2, and again between 3 and 4a and 3 and 4b. Keep this simple, use the same wedges and same glacio-fluvial sediments. Truncate the wedges and show any rejuvination.*

*I suggest starting off with a very obvious channel in the sands/gravels in "a "which should otherwise be pretty level. This deeper channel will fill in with comparatively more silt than the rest of the fluvial plain, but you can show a remnant depression at the surface. The relatively thicker silt/peat allows for more consolidation with thawing, and will give you your deeper central basin.*

We re-drafted the figure, starting with a more obvious channel in the glacio-fluvial sands/gravels in 'a'. We also kept the same morphology of ice wedges and the same thickness of glacio-fluvial sediments from frame to frame.

*P33, Fig. 6a. The ice wedges are not truncated by the active layer and they should be.*

We modified the figure.

*P33, Fig. 6c. Where did the gravel and sand go? Unless there is some sort of new process to remove sediment at depth, this seems odd.*

We corrected the thickness of this unit, so now it is the same as in 'a' and 'b'.

*P33, Fig. 6c. More the arrow to beneath the thaw front and label horizontally. This will help the reader to understand that you mean for thaw consolidation to take place in the silt/peat.*

We modified the figure as requested.

*P33, Fig. 6d. Isn't the lake depth really greater than 4 m? This is what you show.*

We fine-tuned the figure, so that now maximum lake depth is around 4 m (4.2 m).

*P33, Fig. 6d. Little to no subsidence. Sands and gravels are already consolidated and ice poor. Delete.*

We removed subsidence in 6d.

*P33, Fig. 6f. The jump from Stage 3 to 4b is a jump. 4 b implies much deeper thaw beneath the bottom-fast ice than you portray even in 4b, let alone 3, and with drainage one might expect a thaw depth that is similar in depth to stage 0, or at least less than what is shown in 4a. You could show ice-wedge rejuvenation if you showed a shallower thaw depth in 4b than in 3.*

Chronologically, stage 4b would happen notably later compared to stage 4a (if occurring). We mention this in the text (i.e. lake partial infilling followed by partial drainage) (lines #434 to 466). We also modified thaw depth in frame f (stage 4b), so that it is now comparable to frame a (stage 0), i.e. closer to the surface.

*P33, Fig. 6f. On frame "a" this dashed line is called the active layer. How can you have an active layer underneath the lake? May be just call it the thaw front?*

We changed for 'thaw front' when it is underneath the lake (see frame d – stage 3).

*P33, Fig. 6f. I cannot understand why you have invoked frost heave in the sands and gravels too. The sands and gravels should not change from panel to panel unless there is erosion, but the model always has other sediment on top.*

We invoke frost heave in the silt-peat unit only, not in the sands and gravels. There is no erosion of the sands and gravel, we re-drafted the figure to make it more clear.

*P33, Fig. 6f. Delete this markup and add in a comment about whoever in the Department drafted it to the acknowledgments if necessary.*

We deleted the markup and mention the name of the person who drafted it in the Acknowledgments section.

[revised manuscript text omitted]

---

## Editor Decision (ED3)

[revised manuscript text omitted]

The following are review/markup comments visible on the figure:

- This set of figures still needs work. A problem with this set of figures is that the base figure, "a", is not propagated through the rest of the model. The wedges sometimes change from frame to frame (notwithstanding truncation ), and the glacio-fluvial sediments change from Stage 1 to 2, adn again between 3 and 4a and 3 and 4b. Keep this simple, use the same wedges and same glacio-fluvial sediments. Truncate the wedges and show any rejuvination.

  I suggest starting off with a very obvious channel in the sands/gravels in "a "which should otherwise be pretty level .This deeper channel will fill in with comparatively more silt than the rest of the fluvial plain, but you can show a remnant depression at the surface. The relatively thicker silt/peat allows for more consolidation with thawing, and will give you your deeper central basin.

- The ice wedges are not truncated by the active layer and they should be.

- Isn't the lake depth really greater than 4 m? This is what you show.

- Little to no subsidence. Sands and gravels are already consolidated and ice poor. Delete.

- The jump from Stage 3 to 4b is a jump. 4 b implies much deeper thaw beneath the bottom-fast ice than you portray even in 4b, let alone 3, and with drainage one might expect a thaw depth that is similar in depth to stage 0, or at least less than what is shown in 4a. You could show ice-wedge rejuvenation if you showed a shallower thaw depth in 4b than in 3..

- On frame "a" this dashed line is called the active layer. How can you have an active layer underneath the lake? May be just call it the thaw front?

- More the arrow to beneath the thaw front and label horizontally. This will help the reader to understand that you mean for thaw consolidation to take place in the silt/peat.

- I can not understand why you have invoked frost heave in the sands and gravels too. The sands and gravels should not change from panel to panel unless there is erosion, but the model always has other sediment on top.

- Delete this markup and add in a comment about whoever in the Department drafted it to the acknowledgments if necessary.

Figure labels visible: (2100 BP). Ice Wedge. Silt/Peat. Glacio-Fluvial (Sand/Gravel). Marine Silt/Clay. 3: Present-day conditions - Thermokarst lake development (lake depth > 2 m at center). Ice Cover. THER. Gyttja. SUBSIDENCE. inception, ice-wedge melting. Pond. e) Stage 4a: Future development - Lake sediment infilling (gyttja). SEDIMENTATION. nce, lake depth < 2 m. Coalescent pond. SUBSIDENCE +. f) Stage 4b: Future development - Partial lake d. FROST HEAVE. Department of Geography, Lava.

**Figure 6: Four-stage conceptual model of thermo...** [evol]ution through the Holocene. a) conditions with networks of ice wedges develope[d in] [gl]acio-fluvial sand and gravel (and [likely reaching] underlying marine silts and clays). A pre-existing topographic depression of 1-2 m was collecting drifting snow and meltwater. b) Stage 1: thermokarst inception, i.e. deepening of the active layer, melting of the top of ice wedges (triggering ice wedge truncation), development of a hummocky surface. c) Stage 2: thermokarst pond coalescence, formation of a small lake with a maximum depth still above maximum ice cover thickness. d) Stage 3: thermokarst lake mature development by lateral expansion (thermal and mechanical erosion) and bottom deepening (subsidence). Lake maximum depth is now below maximum ice cover thickness, triggering the formation of a talik. e) Stage 4a: possible future evolution by lake infilling (gyttja accumulation). f) Stage 4b: possible future evolution by lake drainage (partial or complete) and re-activation of ice wedge cracking and growth (i.e., no more truncation).

[Figure]

**Figure 7: Signs of past partial drainage around "Gull Lake 2" (GL-2). An inlet flowing from Gull Lake, located south, and an outlet draining towards the nearby proglacial river, are shown in blue. Former lake shores are shown by the dashed red lines. Pingos are indicated by a 'P'. Satellite image: GeoEye-1, 18 July 2010.**